# Mammalian cell growth characterisation by a non-invasive plate reader assay

Alice Grob ⓘ [1,2,7], Chiara Enrico Bena ⓘ [3,6,7], Roberto Di Blasi ⓘ [1,2], Daniele Pessina[1], Matthew Sood[1], Zhou Yunyue[4], Carla Bosia[3,5] ✉, Mark Isalan ⓘ [2,4] ✉ & Francesca Ceroni ⓘ [1,2] ✉

Automated and non-invasive mammalian cell analysis is currently lagging behind due to a lack of methods suitable for a variety of cell lines and applications. Here, we report the development of a high throughput non-invasive method for tracking mammalian cell growth and performance based on plate reader measurements. We show the method to be suitable for both suspension and adhesion cell lines, and we demonstrate it can be adopted when cells are grown under different environmental conditions. We establish that the method is suitable to inform on effective drug treatments to be used depending on the cell line considered, and that it can support characterisation of engineered mammalian cells over time. This work provides the scientific community with an innovative approach to mammalian cell screening, also contributing to the current efforts towards high throughput and automated mammalian cell engineering.

Recent advances in technology and in the automation of cell screening and engineering have made the design and assembly of large libraries of genetic DNA systems possible in mammalian cells[1–3]. This progress builds upon experimental pipelines that have been established to enable the automation of design-build-test analysis of genetic designs in microbes[4,5]. However, the fast identification of desired variants for various applications is only possible if the behaviour of each construct is assessed in the cellular host. It is important to characterise stability, productivity and performance over time, and a key requirement is to measure the impact on cell growth of different constructs.

Whereas high-throughput growth-tracking methods have been developed for bacterial cells[6], such as simple measurements based on light absorbance[7,8], automated and non-invasive characterisation of both engineered and non-engineered mammalian cells is lagging behind.

Traditional methods for mammalian cell growth characterisation include trypan blue staining, flow-cytometry[9], automated cell counters[10,11] and colorimetric assays[12]. However, these methods are disruptive as they rely on a sample being measured and as such only give a fixed sample measurement in time. Also, they are low throughput and time-consuming, representing a limiting step in the development of high throughput approaches. Protocols that have been developed for indirect counting of mammalian cells include digital holography[13], microscopy[14–17], confluency analysis based on commercially available plate readers[17], adoption of magneto sensors[18] and identification of cell-specific optical density (OD)[19,20]. These are mostly restricted to adherent, monolayered or coated cells, can be expensive and can be limited by the number of samples that one can screen at any one time. The implementation of continuous measurements for both adherent and suspension cells would thus allow for a universal, dynamic, and automated mammalian cell characterisation approach.

To advance the experimental characterisation of mammalian cells, we developed a method for non-invasive, high throughput and automated tracking of mammalian cell growth based on the change in absorbance of the pH indicator phenol red (Fig. 1a). This colour indicator is present in a variety of commonly adopted mammalian growth

[1]Department of Chemical Engineering, Imperial College London, London, UK. [2]Imperial College Centre for Synthetic Biology, Imperial College London, London, UK. [3]Italian Institute for Genomic Medicine, Torino, Italy. [4]Department of Life Sciences, Imperial College London, London, United Kingdom. [5]Department of Applied Science and Technology, Politecnico di Torino, Torino, Italy. [6]Present address: Université Paris-Saclay (INRAE), AgroParisTech, Micalis Institute, 78350 Jouy-en-Josas, France. [7]These authors contributed equally: Alice Grob, Chiara Enrico Bena. ✉e-mail: carla.bosia@polito.it; m.isalan@imperial.ac.uk; f.ceroni@imperial.ac.uk

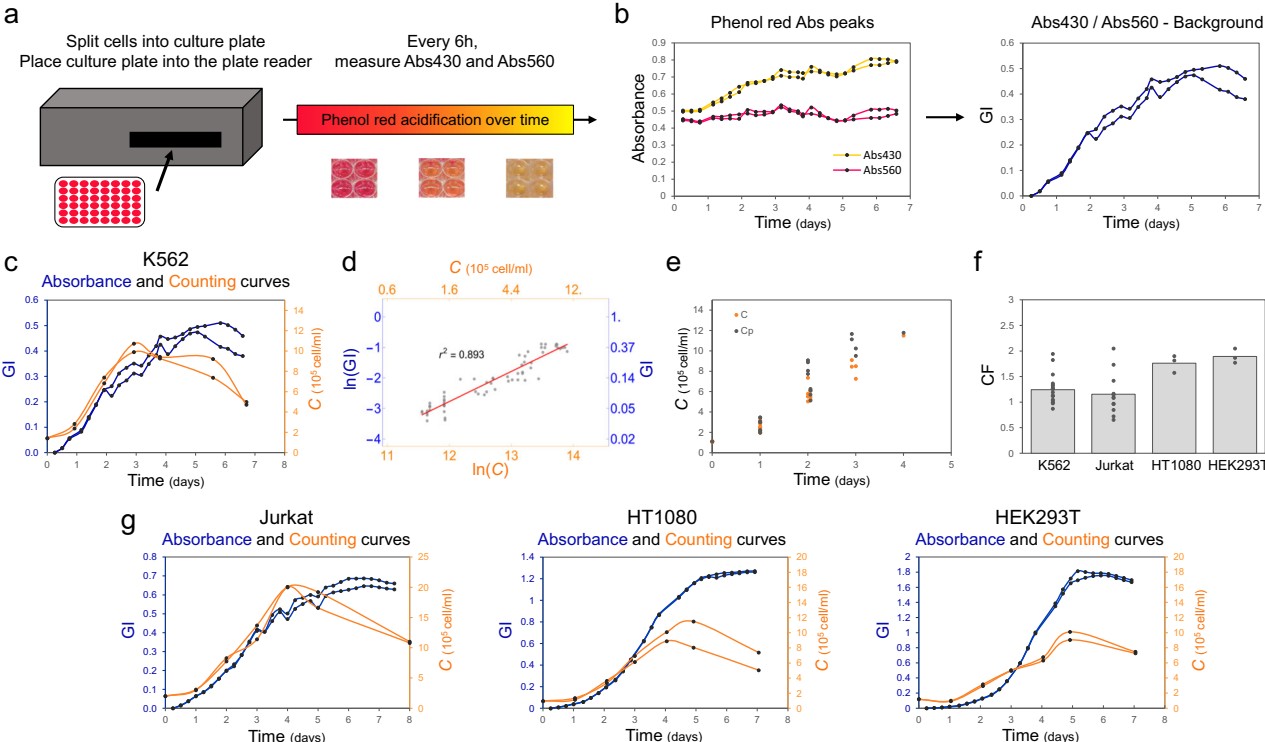

**Fig. 1 | A plate-reader assay based on the absorbance of the pH indicator phenol red can be used to dynamically characterise the growth of mammalian cell lines. a** Schematic of phenol red-based mammalian growth assay. **b** Representative output of the assay with dynamic of $Abs_{430}$ (yellow) and $Abs_{560}$ (pink) (left). The resulting ratio of $Abs_{430}$ over $Abs_{560}$ normalised to the background yields the growth index (GI) profile over time for K562 cells (right). **c** Representative growth curves resulting from phenol red acidification (GI, blue, left vertical axis) and cell counts (C, orange, right vertical axis) of K562 cells. All replicates can be found in Supplementary data file 1. **d** Linear relation between K562 cells ln(GI) and ln(C). The red line is the best fit of data within the linear region, i.e. ln(GI) = 0.99*ln(C) − 14.6 with a coefficient of determination $r^2$ = 0.893. **e** The data set of GI measurements obtained from K562 cells was split. One subset

was used to establish the linear relation between ln(GI) and ln(C) while in the other subset C was predicted from GI measurement according to the fit shown in panel (**d**), i.e. C = (GI/exp(q))^(m). The resulting predicted C ($C_p$, grey) is plotted here with the measured C (C, orange) as a function of time. **f** Bar plot of conversion factors (CF) for suspension cells (K562 and Jurkat) and adherent cells (HT1080 and HEK293T). The height of the bar represents the mean value of the single replicates shown as black dots. **g** Representative profiles of GI (blue, left vertical axis) and C (orange, right vertical axis) for Jurkat, HT1080 and HEK293T cells. All repeats can be found in Supplementary Data File 1. All CF values can be found in Supplementary Data File 2. The numbers of biological repeats for each sample are reported in Table S3. Data analysis is described in the Methods section and in Supplementary Note 1. Source data are provided as a Source Data file.

media and is available separately as a dye that can be added to any medium of interest[21]. The acidic and basic forms of phenol red are characterised by absorbance peaks at 560 nm and 430 nm ($Abs_{560}$ and $Abs_{430}$) respectively[22,23]. During mammalian cell growth, phenol red gradually transitions from its basic to its acid form, providing the operator with a quick and visual indication that cells are growing and reaching confluence. Previous work showed that the change in absorbance of phenol red can be used as a valuable method for tracking cell growth in bacteria[24,25] but no one has so far provided a method for mammalian cell tracking using the same principle. Our workflow enables characterisation of mammalian cell growth over time, and we demonstrate this working for both suspension and adhesion cell lines, a clear step forward compared to current state of the art. We also confirm that this method can be adopted for a wide range of applications from routine mammalian cell analysis to a screening of cell sensitivity to drug treatment, but also for the advancement of high throughput characterisation of engineered constructs in these cells, thus making it relevant for both basic and applied research.

## Results

### A phenol red-based growth assay enables plate reader tracking of mammalian cell growth

We started by considering the suspension cell line K562[26] and designed a routine protocol where cells are first seeded at their recommended

seeding density in a 48-well plate. The plate is then inserted into a plate reader where the temperature is set at 37°C and $CO_2$ at 5%. Absorbance readings for the basic ($Abs_{560}$) and acidic ($Abs_{430}$) forms of phenol red are performed every six hours for up to nine days (Fig. 1a).

It was previously suggested for microbes, that a growth index (GI) can be identified by the ratio between $Abs_{430}$ and $Abs_{560}$[24,25], which yields a characteristic sigmoidal growth profile (Fig. 1b, Fig. S1a−c). To confirm that a GI is also a reliable proxy of cellular growth in mammalian cells, we performed parallel cell counts every 24 hours for each sample, using a standard automated cell counter (see Methods section). We reasoned that this would allow us to compare the growth profiles yielded by the two methods and to benchmark against one of the techniques most adopted for following mammalian cell growth. As expected for batch cell cultures, cell concentration (C) yielded a typical sigmoidal curve, inclusive of lag, exponential and stationary phases of growth (Fig. 1c). Plate reader growth curves showed an analogous growth profile (Fig. 1c, Supplementary data file 1).

Specifically, we identified a range for which the logarithm of C (ln(C)) and the logarithm of GI (ln(GI)) follow a linear relation and that this corresponds to the exponential phase of growth (Fig. 1d, red line, Table S1 and Supplementary Note 1). The slope of the linear fit between ln(GI) and ln(C) provides a calibration curve to establish the relation between the actual number of cells and the detected absorbance ratio (ln(GI) = $m$*ln(C) + $q$, Table S2). For K562 cells, a slope value close to 1 (i.e. 0.99) suggests that GI scales almost

linearly with C (see Table S2 and Supplementary Note 1). Moreover, from the relation existing between ln(GI) and ln($C$), $C$ can be estimated from GI as $(GI/exp(q))^{(1/m)}$ (Supplementary Note 1). Thus, to fully benchmark our method against cell counts, we decided to use a data subset to establish the relation between $C$ and GI and use it to predict $C$ ($C_p$) from GI in the remaining data subset in exponential phase (Fig. 1e). $C_p$ values overlap with actual $C$ values over time, confirming the relation existing between ln(GI) and ln($C$) in exponential phase.

Thanks to this relation, it was also possible to compute the growth rate from plate reader ($\mu_p$) and cell counts ($\mu_c$) in the same way, i.e. by computing the slope of the data within the exponential phase as a function of time (Supplementary data file 2 and Supplementary Note 1 for data analysis details). In order to establish the relationship between $\mu_p$ and $\mu_c$, we developed a protocol for automated analysis that enabled the calculation of the growth rate from each dataset (see Supplementary Note 1 and Supplementary software file). We named the conversion factor (CF) the ratio between $\mu_p$ and $\mu_c$. We found that CF for K562 cells is around 1, suggesting that the indirect $\mu_p$ computed through GI is a proxy for the effective $\mu_c$ of the population obtained by direct cell counting (Fig. 1f, Supplementary data file 2 and Supplementary Note 2). This relation is valid within the exponential phase, when $\mu_c$ and $\mu_p$ scale linearly to each other, as shown in Fig. 1d. Analogously to the relation observed in bacteria between OD and cell concentration[8], linearity between the two quantities is lost when cells start saturating. As evident from Fig. 1c, after saturation, the growth curve for $C$ decreases more rapidly than the one for GI.

Taken together these results demonstrate that a phenol red-based plate reader assay allows tracking of mammalian cells growth in standard conditions and that $C_p$ and $\mu_p$ computed through GI are accurate proxies for $C$ and $\mu_c$ respectively.

### Suspension and adhesion cell lines can be characterised by plate reader-based analysis

We next set out to verify whether it was possible to apply this method to a range of cell lines, and specifically to both suspension and adherent cells. We thus adapted our protocol to characterise the growth of a second suspension cell line (Jurkat[27]) and two cell lines growing in adhesion (HT1080[28] and HEK293T[29]). As shown in Fig. 1f, when the cells are grown in standard growth medium at 37 °C, GI captures the trend of the growth profiles of all cell types reliably (Fig. 1g, Fig. S1d–f, Fig. S2), again leading to a linear relation between ln(GI) and ln($C$), within the exponential phase of growth (Fig. S3, Tables S1 and S2). When we compared the CF for the four different cell lines, we noticed that the CF for Jurkat cells is also close to 1, as for K562 cells, while both adhesion cell lines show an average CF closer to 2 (Fig. 1f, Table S4 and Supplementary Note 1). To explain this difference, we considered that both K562 and Jurkat cells are cultured in RPMI medium (containing 5 mg/L phenol red), while HT1080 and HEK293T cells are cultured in DMEM (containing 15 mg/L phenol red). Indeed, we noticed that $Abs_{560}$ follows a very different trend for cells grown in DMEM (Fig. S2a, b), compared to RPMI (Fig. S1a, b), with a more pronounced decreased over time in the former case. This is then reflected in a wider GI variation (Fig. S2e, f). It was previously suggested that DMEM, and specifically $Abs_{560}$ in this medium, is more sensitive to pH change if compared to RPMI, leading to the hypothesis that the difference in CF is directly linked to a medium-dependent effect that leads to a different profile in $Abs_{560}$[30].

To test this experimentally, we purchased DMEM where no phenol red is present and considered HT1080 cells growing in DMEM supplemented with 15 mg/L phenol red for reference, as in the original DMEM formulation adopted in the experiments presented in this manuscript and in DMEM supplemented with 5 mg/L of phenol red, same concentration present in other commercially available media, like the RPMI adopted to grow the suspension cell lines of this study.

Cell growth was not affected by phenol red concentration as displayed by the cell counts (Fig. S4a). When considering GI, we confirmed that in the presence of 15 mg/L phenol red, GI saturates at a higher value and with a higher average growth rate when compared to the case of 5 mg/L (Fig. S4b–e). Linearity is not impacted ($r^2 = 0.91$ for 15 mg/l and 0.95 for 5 mg/l) and a corresponding CF can be identified, similar to what previously shown in Fig. 1e (Fig. S4f, g). CF for HT1080 growing with 15 mg/L phenol red is consistent with the one previously obtained in Fig. 1e. For HT1080 growing with 5 mg/L phenol red, the average CF drops to a range compatible with what previously observed for suspension cell lines grown in RPMI. This additional data set suggests that CF may indeed depend on the experimental condition, and possibly the cell line, adopted in the assay. Nonetheless, it also confirms that a linear relation can be found between ln($GI$) and ln($C$) when the cell line is initially characterised.

Overall, we confirmed that, like other methods routinely adopted in cell and molecular biology (e.g. bacterial OD[8], protein quantification), the workflow requires an initial calibration curve for identification of the linear relation between ln(GI) and ln($C$). Once this is obtained, $\mu_p$ can be treated and considered as a direct proxy for the actual cellular $\mu_c$, making the method generalisable for both suspension and adhesion cells.

### Phenol red-based growth assay enables mammalian growth tracking under different conditions

Once we established that our method could reliably follow the growth profiles of different cell types, we investigated if it could be adopted for the characterisation of cell growth across different culture conditions. Temperature and carbon source are two key parameters in mammalian process optimisation, and they have been previously shown to impact bioproduction and therapeutic applications[31–33]. Proving that our protocol is robust to varying environmental conditions is thus an essential requirement to confirm it can be applied to wider mammalian cell analysis.

We focused on comparing K562 and Jurkat cells growing at 37 °C and 33 °C, in the presence of glucose (Glu) or mannose (Man) (Fig. 2 and Fig. S5). GI profiles still reliably captured cell count profiles, for both cell lines growing at 33 °C (Fig. 2a and S5a) and when Man was used as carbon source (Fig. 2b, c, S5b and c). Data analysis by our automated pipeline confirmed that changing growth conditions did not affect the linear relation previously observed between ln(GI) and ln($C$) (Fig. 2d–f and S5d–f). Importantly, the average value of CF for both cell lines did not change (Fig. S6a, Table S5 and Supplementary data file 2). Based on these results, we decided to verify once again how automatable our pipeline is and verify if, once we established GI, $\mu_p$ and CF for a given cell line and condition, it could be possible to infer $\mu_c$ for subsequent experiments where only plate reader measures of GI are performed in the same conditions. We thus used the calculated $\mu_p$ and CF from an initial set of measures for K562 and Jurkat cells growing at 37 °C and 33 °C, in the presence of Glu or Man, to estimate $\mu_c$ for subsequent plate reader runs and compared the estimated values ($\mu_{cp}$) with the actual $\mu_c$ calculated by parallel cell counting (Fig. S6b). $\mu_{cp}$ closely matched the measured $\mu_c$, thus confirming that once an initial characterisation is performed, the identified CF can be adopted to infer information on actual $\mu_c$ of the cells in further experiments without the need for cell counting.

### High throughput screening of therapeutic drugs enables identification of cell line-specific response

Doxorubicin (doxo) is a well-known chemotherapeutic agent adopted in anticancer medications to treat, amongst others, breast cancer and leukaemia[34]. Doxo inhibits cell growth via interference with DNA replication and RNA synthesis, leading to cell death[35]. Here, we set to use doxo as testbed to investigate the suitability of our workflow for testing the effect of drug treatment on mammalian cells. Indeed, while

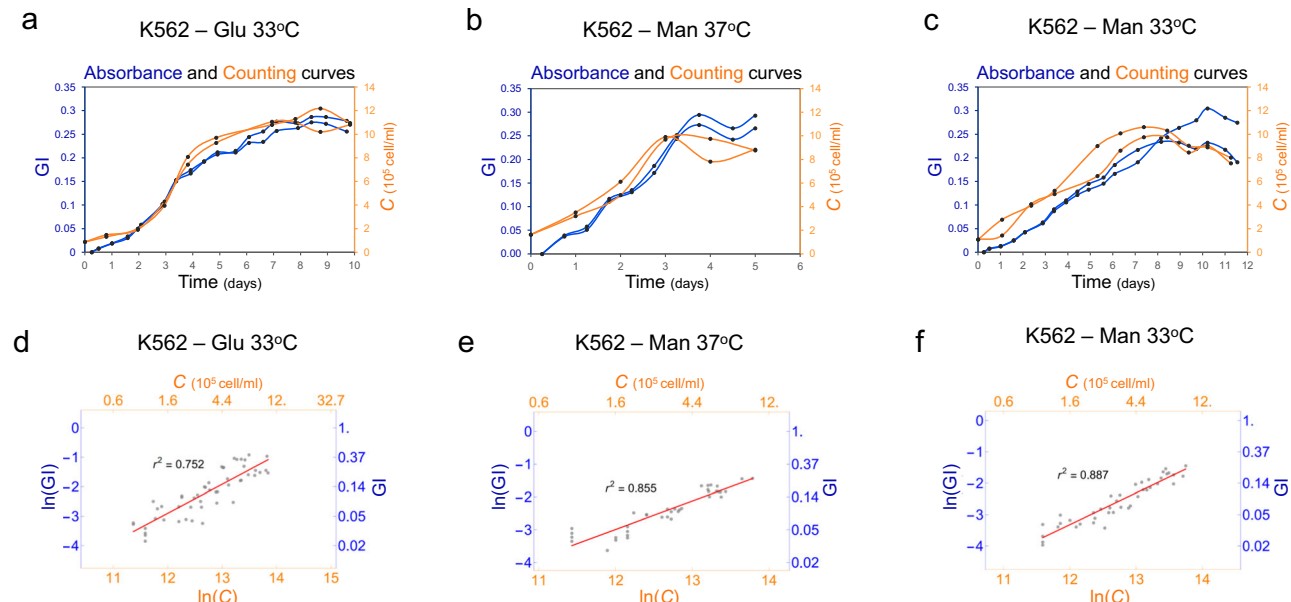

**Fig. 2 | Growth of mammalian cell lines under different conditions can be characterised by a plate reader assay.** Representative growth index (GI, blue, left vertical axis) and cell count (*C*, orange, right vertical axis) profiles over time of K562 cells grown with glucose (Glu) **a** at 33°C or mannose (Man) at either **b** 37°C or **c** 33°C. All replicates can be found in Supplementary data file 1. Linear relation for ln(GI) and ln(*C*) of K562 cells grown with Glu **d** at 33°C or Man at either **e** 37°C or **f** 33°C, indicating r² values of 0.752, 0.855 and 0.887 in exponential phase, respectively. Red solid lines correspond to the linear fit. Fit equations are reported in Table S2. The numbers of biological repeats for each sample are reported in Table S3. Data analysis is described in the Methods section and in Supplementary Note 1. Source data are provided as a Source Data file.

traditional methods for identification of effective doxo concentrations, and for screening sensitivity of different cell lines to the drug, involve manual or automated cell counting, we reasoned that our platform could provide an improved workflow for high throughput testing.

To investigate this, we first verified that the addition of chemicals to the culture medium does not affect GI over time. For this purpose, we supplemented RPMI and DMEM with doxo, as well as with other commonly adopted inducer molecules (doxycycline, dox; and IPTG), antibiotics (blasticidin, blast; puromycin, puro; and hygromycin B, HygB) and the drug solvent DMSO. The addition of these compounds had no impact on medium acidification over time as shown in Fig. S7.

We then moved forward to apply our protocol to test the effect of different doxo concentrations on two of the four cell lines considered, namely HT1080 (known to be doxo sensitive[36,37]), and K562, selected as testbed of a leukaemia cell line[26]. Both cell lines were treated with 0, 10, 25, 50, 75, 100, 500, 750 nM and 1uM Doxo, added at 24 hours after the start of the assay, and parallel GI measurements and cell counting were performed (Fig. 3a, b). We confirmed that $\mu_p$ and $\mu_c$ displayed a similar decreasing trend for increasing doxo concentrations in both cell lines as shown by the regression lines in Fig. 3c, d with angular coefficients of −15.887 and −16.414 for $\mu_p$ and $\mu_c$ in HT1080, respectively, and −20.212 and −19.296 for $\mu_p$ and $\mu_c$ in K562, respectively. Furthermore, it as possible to identify a specific cell line response to doxo treatment. Indeed, HT1080 cells treated with 75 nM doxo resulted in an average 30% decrease of $\mu_p$ and 84% decrease of $\mu_c$, whereas K562 cells displayed less sensitivity to the same doxo treatment with an average 17% decrease of $\mu_p$ and 33% decrease of $\mu_c$ compared to the untreated control.

Finally, while analysing the data, we noticed that, for increasing concentrations of doxo, both GI and cell count curves flatten, as a reflection of cell death (Fig. 3 and Supplementary data file 1). This led to a more difficult estimation of $\mu_p$ and $\mu_c$, due to a less pronounced exponential phase of growth needed to estimate both the linearity region and growth rates (see Supplementary Note 1). As previously shown for bacteria, treatment with molecules that affect cell viability may impact such linearity[8]. This needs to be considered as it may lead to the case where the assay is qualitatively very useful but is only semi-quantitative.

To corroborate these results, we considered Jurkat cells treated with the microtubule inhibitor colchicine, an inhibitor of cell division that, similarly to doxo, impacts cell growth[38]. GI measurement and parallel counting were performed for cells treated and untreated with 0.025ug/ml colchicine 24 hours after the start of the assay (Fig. S8). Results showed that after colchicine addition, $\mu_c$ decreased by -100% in treated cells compared to cell growth rate before treatment, while $\mu_p$ decreased by -75% (Fig. S8). This reflective change of $\mu_p$ indicated that GI curves can again capture changes in growth rate occurring over the duration of the assay thus mirroring the cell counts. To conclude, while our proposed assay is mainly suitable for actively growing cells for which a linearity window can easily be identified, it still can be adopted for semi-quantitative characterisation of the dose-response effect of drug treatments on mammalian cell viability with a throughput difficult to achieve with standard cell counting approaches.

## Automated tracking of engineered mammalian cells and single cell construct performance

Finally, we thought to apply our method to the characterisation of engineered mammalian cells, considering that high throughput and automated analysis of engineered hosts is much needed within the biotechnology and synthetic biology communities. To provide proof-of-concept of the suitability of our method for this task, the HEK293TLP-EBFP cell line, a HEK293T cell line bearing a landing pad (LP) cassette integrated in the AAVS1 locus (available from Matreyek et al.[39]) was selected. The LP cassette codes for an EBFP expressed under an inducible Tet promoter. The LP also bears a second transcriptional unit expressing the rtTA transactivator under the control of a constitutive EF1α promoter (Fig. 4a). When doxycycline (dox) is present, the rtTA transcription factor activates the Tet promoter and EBFP is expressed. Conversely, when dox is absent, the cassette is off. Since

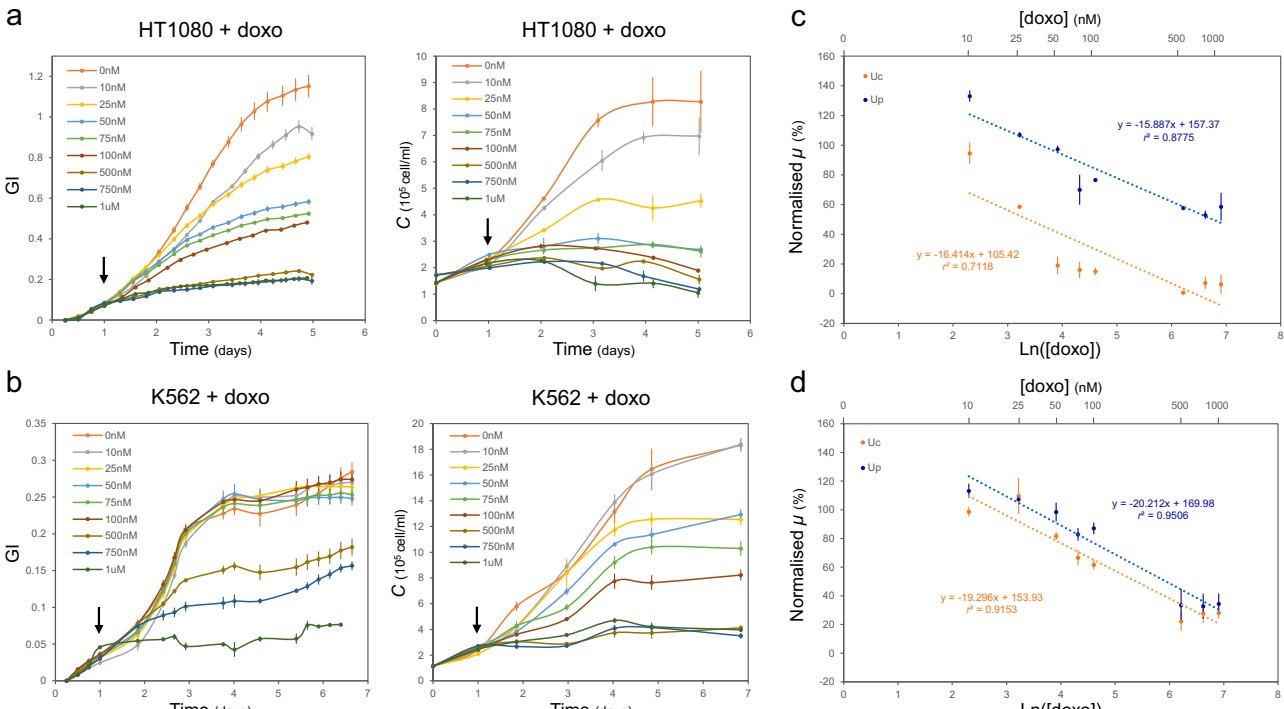

**Fig. 3 | Growth of mammalian cell lines treated with doxorubicin can be characterised by a plate reader assay.** Response of HT1080 (**a**) and K562 (**b**) cells to treatment with increasing doxorubicin (doxo) concentrations (0 nM to 1 μM). Doxo was added 24 hours after the start of the assay (arrow). Relative growth rates form plate reader assay ($\mu_p$, blue) and cell counts ($\mu_c$, orange) of HT1080 (**c**) and K562 (**d**) cells treated with 0nM-1μM doxo (axis in log scale) were normalized to the average value of $\mu_p$ and $\mu_c$ for cells treated with 0 nM doxo. Dashed blue and orange lines indicate the variation trend of $\mu_p$ and $\mu_c$ respectively for cells treated with 10 nM to 1 μM doxo. Equation and $r^2$ values of these trendline are indicated. Number of biological repeats for each sample are reported in Table S3. Data are presented as mean values +/- SEM. Data analysis is described in the methods section and in Supplementary Note 1. All repeats can be found in Supplementary data file 1. Source data are provided as a Source Data file.

the switch is known to be sensitive to different dox levels, we reasoned it could work as a good proxy for genomic constructs with different expression strengths, thus providing a good testbed for our method.

Firstly, we characterised the HEK293TLP-EBFP cell line by following our protocol as outlined in Fig. 1a (Fig. S9a−c). Performing plate reader measurements and cell counts in parallel (Fig. S9d), we confirmed linearity between ln(GI) and ln($C$) for this additional cell line (Fig. S9e) and established its specific CF (Fig. S9f).

We then reasoned that the presence of phenol red in the medium could impact fluorescence readout. Thus, before assessing the fluorescence response of the cell line, we assessed if crosstalk could be detected between phenol red and fluorescence readings. In order to do so, we expanded the analysis to a second cell line, HEK293TLP-mCherry, developed in-house. Similarly to HEK293LP-EBFP, HEK293TLP-mCherry codes for a mCherry protein under the control of dox induction (see EBFP diagram in Fig. S4a and Source data for sequence). By assessing fluorescence per cell in the presence and absence of phenol red, we confirmed that the presence of phenol red in the medium does not impact the fluorescence readout (Fig. S10).

Once the suitability of the workflow was confirmed, we further characterised the response of the HEK293LP-EBFP cell line to dox induction. To do so, cells were seeded in a 48-well plate and were allowed to grow in the plate reader for one week after induction at T0 with different concentrations of dox. GI for the different samples and total EBFP fluorescence were captured (Fig. 4b, c). As expected, increased dox induction led to increasing EBFP levels.

Bacterial OD is routinely used to infer normalised protein expression per cell. This is essential practice to compare samples and conditions, as it enables us to take differences in growth and in cell number into account when characterisation of a construct

functionality is sought. Thus, we thought to calculate the $C_p$ value corresponding to a given GI value (Fig. 4d), similarly to what we previously did in Fig. 1e (see Supplementary Note 1 for details). We then compared how accurate the prediction of $C$ from GI is compared to bacterial OD. We compared the mean $C_p$ by our calibration method to the mean measured $C$ which reduces to the accuracy defined as $A = C_p/C$. The closer $A$ is to 1, the more accurate the method is in estimating cell counts. We identified a similar accuracy in our approach compared to the accuracy of OD measurements. Indeed, for K562 cells, $A$ equals (1.07 +/− 0.02), in agreement with the accuracy defined for bacterial cells[40]. The procedure and results of the accuracy calculations can be found in Supplementary Note 3. Once we had confirmed the accuracy of our approach, we adopted the calculate $C_p$ for the HEK293TLP-EBFP cell assays to normalise the protein expression on the estimated number of cells at a given moment in time (i.e. three days), within the linearity window, and obtain an estimated quantitative measure of expression levels per cell (Fig. 4e). In the work published by Matreyek et al., authors performed flow cytometry measurements and characterisation of the induction dynamic of the HEK293LP-EBFP cell line, looking at EBFP signal expression over time. They monitored that cells induced with 2 ng/μl dox displayed an increase in EBFP expression per cell up to five days. Our normalised EBFP data (Fig. S11) show a similar increase in EBFP per cell over time after induction. In conclusion, our method led us to similar single-cell construct characterisation but via a non-invasive and high throughput method that enabled both growth and fluorescence tracking with no requirement for manual handling.

## Discussion

The ability to track mammalian cell growth and performance over time, adopting automated and non-invasive protocols, is key to

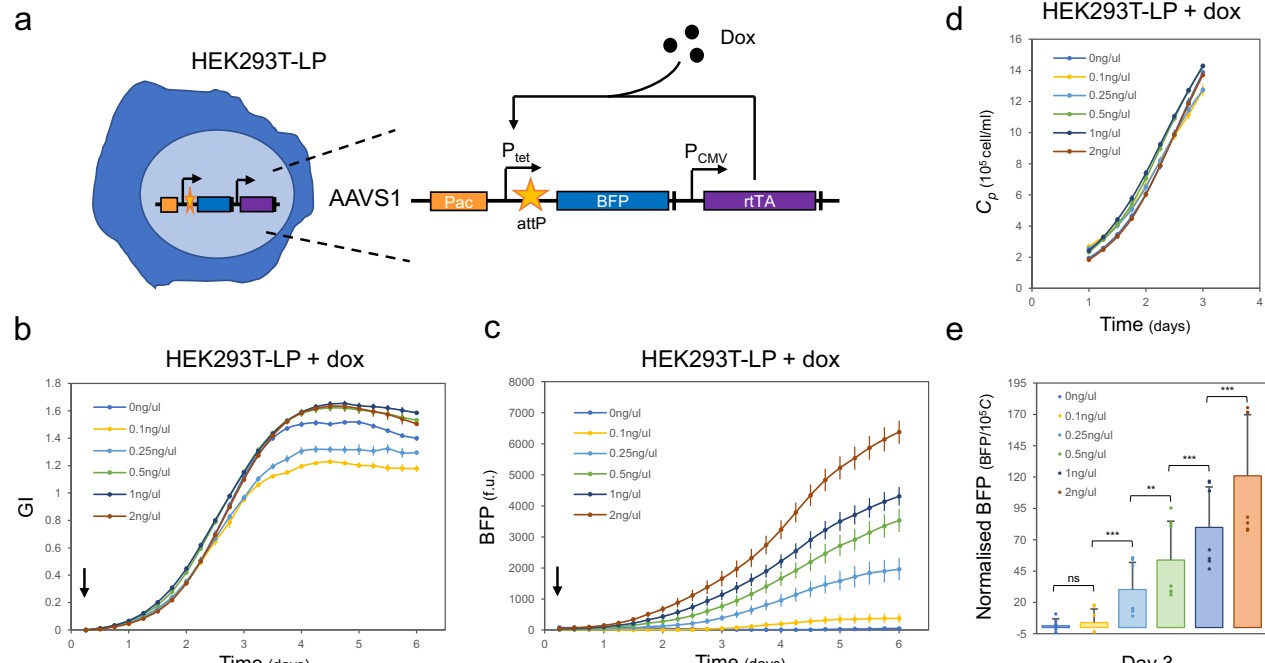

**Fig. 4 | Gene expression from a landing pad cassette integrated in HEK293T cells can be characterised over time with a plate reader assay.** **a** Schematic of the landing pad (LP) in HEK293TLP-EBFP cell line from Matreyek et al.[39]. A cassette expressing the dox-responsive rtTA transactivator and the inducible Ptet-EBFP transcriptional unit is integrated into the AAVS1 locus of HEK293T cells. **b** growth index (GI) and **c** EBFP fluorescence of HEK293LP-EBFP cells can be followed over a one-week time frame for samples induced with increasing concentrations of dox (0 ng/μl to 2 ng/μl). The addition of dox at T0 is indicated by an arrow. **d** Estimated $C$ ($C_p$), derived from $GI$, within the linear time frame between the two variables (i.e. one to three days). **e** Bar plot of fluorescence per cell, calculated by normalising the total EBFP fluorescence on the estimated $C$, at three days post induction. Data are presented as mean values +/- SD with single replicates shown as dots. A two-sided t-test indicated that normalised EBFP levels increase significantly in cells treated with 0.25 ng/μl dox and above (ns, non-significant: $p > 0.5$, **: $p < 0.01$ and ***: $p < 0.001$). Numbers of biological repeats for each sample are reported in Table S3. Data analysis is described in the Methods section and in Supplementary Note 1. Exact p values and source data are provided as a Source Data file.

support current advances in mammalian cell research. In this study, we developed a first-of-its-kind plate-based high throughput method for the characterisation of mammalian cell growth. The method is based on the change in absorbance observed for the colour indicator phenol red during growth in mammalian cultures. It consists of four main steps, in which (i) cells are first seeded with a desired concentration and growth medium into a 48-well plate that is then inserted into a plate reader with temperature and $CO_2$ control; (ii) measurements of $Abs_{430}$ and $Abs_{560}$ are performed every six hours for one week; (iii) a ratio of these absorbances is computed in order to generate a GI curve; (iv) automated data analysis is used to identify the growth rate of the GI curve in exponential phase. An initial calibration is required for any new cell line to be analysed, and this can be done by performing cell counts every 24 hours in parallel to the initial plate reader measurements. The workflow then supports analysis of the cell counts and plate reader curves to identify their corresponding growth rates. The resulting ratio between the calculated growth rate in a plate reader, $\mu_p$, and the growth rate from cell counts, $\mu_c$, defines a conversion factor, CF, which holds in exponential phase, i.e. when $\ln(GI)$ and $\ln(C)$ scale linearly. We showed that, for each growth curve, CF is compatible to the slope of the linear relation between $\ln(GI)$ and $\ln(C)$.

We started by showing that our method enables tracking of both suspension and adhesion cell lines, going beyond existing methods currently restricted to one of the two cell types[15,17]. We then showed that together with standard growing conditions, the method can be used to characterise cell growth and performance when temperature and medium composition are changed. The method also successfully provides characterisation of cell-specific responses to chemotherapeutic drug treatment and identification of effective working drug concentrations. However, when chemical agents are adopted that impact cell growth, the linearity between $\ln(GI)$ and $\ln(C)$ may be affected as previously reported for bacterial OD[8]. While this may impact the quantitative power of the approach, we showed that GI still closely follows the trend of cell counts, providing a useful workflow for high throughput analysis of cell viability.

When comparing our approach to OD, it must also be considered that, while OD is based on the measurement of scattered light from cells in solution, GI in our workflow is based on media acidification. Thus, GI is a dynamic measurement of mammalian cell growth dependent on the conditions and history of growth, while OD enables assessment of cell concentration at any given moment in time, independently of the history of the cell population.

Finally, we demonstrated that engineered mammalian cells can be characterised in their growth and fluorescent outputs when an integrated cassette is expressed. This will need to account for an initial screen to ensure no crosstalk is present between a given fluorescent protein and phenol red in the medium, as shown by our data for EBFP and mCherry. Importantly, we provide evidence that, once an initial calibration is established for a given cell line in a specific condition, the method can be adopted to infer actual $\mu_c$ and cell numbers from plate reader measurements alone. By adopting the relation existing between GI and $C$ to convert plate reader data into cell counts, the protocol enables quantification of construct and cell performance over time when different induction is sought. This strengthens the relevance of the methods widening its potential applications in mammalian cell analysis.

Our method has the potential to greatly support advancements in the automation of mammalian cell screening both in basic and applied research. For the latter, while for several micro-organisms it is now possible to perform automated library assembly and parallel testing of

high number of variants[41–43], the uptake of the same workflow in mammalian cells is lagging behind. Design and assembly of large construct libraries is now possible for mammalian systems thanks to the development of toolkits that support modular and high throughput construct generation[44,45], but the screening of such variants within their target host is slowed down by the low throughput protocols that were previously available. Biofoundries have been created over the last decade to address this need, specifically aiming to introduce automation into the well-known design-build-test cycle of synthetic biology[46]. In parallel, important technological advances have arisen, like the ones showcased by companies such as Berkley Lights that allow automated cell line development and clone selection[47,48]. However, such technologies are still currently inaccessible to lab-end users, due to their high costs and need for dedicated lab space.

Methods which are more easily accessible, such as the one developed here, will thus be pivotal in supporting the advancement of mammalian cell analysis, and in enabling a more dynamic, automated and higher throughput characterisation of mammalian cells than was thus far possible.

## Methods

### Cell culture
Suspension cells [K562 (ATCC CCL-243) and Jurkat (ATCC TIB-152)] and adherent cells [HT1080 (ATCC CCL-121), HEK293T (ATCC CRL-3216), HEK293TLP-EBFP (HEK 293T AAVS1 LP, kind gift from D.M. Fowler), HEK293TLP-mCherry (bearing genomic expression of mCherry, generated in-house, see Source data for map and sequence) were cultured respectively in RPMI medium 1640 (Gibco A10491-01) or DMEM medium (Gibco 31966-021) with 10% foetal bovine serum (FBS; Gibco). As per ATCC guidelines K562 and Jurkat cell lines do not require continuous shaking. To assess the effect of phenol red concentration, HT1080 cell were cultured in clear DMEM (Gibco 21063-029) supplemented with 15 mg/L or 5 mg/L phenol red (Sigma). For characterisation of growth in different conditions, suspension and adherent cells were pre-cultured for two weeks in their respective glucose-free medium (Gibco 11879-020) with 10% FBS and 25 mM Man (Sigma). This concentration of Man was also adopted for plate base and cell count measurements.

### Plate reader assay
Suspension and adherent cells were resuspended in fresh medium at $10^5$ cells/ml and plated into a 48-well plate (Greiner). To limit evaporation, phosphate-buffered saline (PBS; Sigma-Aldrich) was added in-between wells and the plate was sealed with parafilm on its long sides, leaving the short sides for $CO_2$ exchange. The plate was then placed in a SPARK or an Infinite M200 Pro microplate reader (Tecan) and incubated at 33 °C or 37 °C ± 0.5 with 5% ± 0.5 CO2 for up to 15 days. Every six hours, the plate of suspension cells was orbitally shaken for 20 seconds at 2.5 mm amplitude, incubated for 5 seconds and then $Abs_{430}$ and $Abs_{560}$ were measured. For adherent cells, $Abs_{430}$ and $Abs_{560}$ were measured every six hours without shaking.

### Cell counting
Cell counting was performed in parallel to the plate reader assay. Every 24 hours, the 48 well plate was taken out of the microplate reader, 5 µl of suspension cells were stained with 0.2% trypan blue (Gibco) and counted using a TC20™ automated cell counter (Bio-Rad). Adherent cells were detached using 0.05% Trypsin-EDTA (Gibco) then inactivated by v/v DMEM medium. Cell counting of these cells was performed using a Nucleocounter NC-250™ (Chemometec). The remaining cells were removed from the well and replaced by PBS to minimise evaporation. Subsequently, the plate was sealed again with parafilm on its long sides and replaced inside the microplate reader prior to restarting the plate reader assay.

### Growth curves
For suspension cell lines, cell counting, and absorbance measurements were performed from the same well, from seeding to saturation. Thus, the growth curves obtained for these cells correspond to measurements over time from a single well. On the contrary, for adherent cells, each time point corresponds to a different well due to the need for trypsinization and detachment of the cells before counting. For both C and GI growth curves each time point thus corresponds to the measurement of a given well at a given time. In parallel to the wells used for counting adhesion cells, two wells were solely measured by plate reader for the entire duration of the experiment. GI values were normalised to their initial values at six hours, $GI_0$ (see Supplementary Note 1) and subsequent first measurements upon restarting the plate reader were discarded.

### Bacterial OD600 accuracy assay
The following protocol was followed to perform the first of the two analysis methods presented in detail in Supplementary Note 3. *E. coli* DH10B cells were grown at 37 °C overnight with aeration in a shaking incubator in 5 ml of M9 medium [M9 salts supplemented with 0.4% casamino acids (MP Biomedical), 0.25 mg/ml thiamine hydrochloride (Sigma), 2 mM MgSO4 (Sigma), 0.1 mM CaCl2 (WR), 0.4% fructose (Sigma)] ($n = 6$). In the morning, 45 µl of each sample was diluted into 1.5 ml of fresh M9 media and grown at 37 °C with shaking for another hour. 100 µl of each sample were then transferred into 6 wells of a 96-well plate (Greiner) at approximately 0.25 OD600. The plate was sealed with a Beath-Easy membrane (Merk), placed in an Infinite M Nano+ microplate reader (Tecan) and incubated at 37 °C with orbital shaking at 432 r.p.m. for five hours, with measurements of OD600 taken every 15 min. Every hour, the plate was taken out, the Breath-Easy membrane was removed and 10 µl of each sample was diluted into 1 ml of PBS for flow analysis. This was performed by using a MACS Quant flow cytometer, providing bacterial cell counts in parallel to OD600 measurements. Following sampling, the plate was placed back into the microplate reader and OD600 measurement resumed.

### Growth rates, conversion factors and relation between ln(GI) and ln(C)
To infer the relation between GI and C, we analysed ln(GI) as a function of its correspondent ln(C) values. As discussed in Supplementary Note 1, the two variables scale linearly during the exponential phase of growth of the population of cells. Therefore, we applied the procedure briefly described here below and detailed in Supplementary Note 1 to quantify: i) the growth rates in the exponential phase, ii) CFs, and iii) the relation between ln(GI) and ln(C).

Briefly, the procedure for data analysis is based on the following points:

1. The C growth profiles, expressed as ln(C) vs time, are analysed and the region of fastest growth, i.e. the exponential phase of growth, is identified automatically (see Table I in Supplementary Note 1). The slope of the linear fit of the selected data is $\mu_c$.

2. We selected the very same time-window in the corresponding growth profile ln(GI) vs time. The slope of this second fit is then $\mu_p$ (see Table II in Supplementary Note 1)

3. Finally, the data selected in previous steps 1) and 2) are fitted to obtain the relation between ln(GI) and ln(C) of the form ln(GI) = m*ln(C) + q, being m and q parameters of the linear fit (see Table III in Supplementary Note 1). From point 3. it follows that GI = $e^{\wedge}(q)$* $C^{\wedge}(m)$ and C = $(GI/\exp(q))^{\wedge}(1/m)$. All data were analysed through custom made scripts in *Wolfram Mathematica* (*version 13.1*). The code used for data analysis, description of it and source data file are available as Supplementary software file. See Supplementary Note 1 for further details.

## Data exclusion

Outliers with GI < $GI_0$ or $C < C_O$, where $C_O$ is the initial concentration of cells and $GI_0$ is GI normalised to zero at six hours were excluded. When cell count was performed in parallel to absorbance measurements, first GI measurements upon restarting the plate reader were discarded. GI growth curves with negative value for GI at six hours were excluded from the analysis.

## Media acidification assay

RPMI and DMEM medium supplemented with 10% FBS were added to a 48-well plate. The following compounds doxo (1uM, Fisher Scientific), dox (2 ng/μl, Sigma), IPTG (1 mM, Sigma), Blast (0.5ug/ml, InvivoGen), Puro (1ug/ml, InvivoGen), HygB (50ug/ml, Sigma) and DMSO (0.1%, Invitrogen) were individually added to both media at T0 before running the plate reader assay for seven days.

## Doxorubicin assay

HT1080 and K562 cells were plated into a 48-well plate at $10^5$ cells/ml. The plate reader assay was started and let run for 24 hours. The plate was then removed from the microplate reader and 0 to 1 μM doxo (Fisher Scientific) was added to the plate. The plate was then placed back into the microplate reader and absorbance measures were performed for six more days. Cell count was performed in parallel every 24 hours.

## Colchicine assay

Jurkat cells were plated into a 48-well plate according to the plate reader assay which was started for 24 hours. The plate was then removed from the microplate reader and cells were treated with 0 or 0.025ug/ml colchicine (Sigma). The plate reader assay was then resumed for six more days with parallel cell count was performed every 24 hours.

## EBFP and mCherry expression assay

To assess for crosstalk between phenol red and fluorescence readings, HEK293TLP-EBFP and HEK293TLP-mCherry cells were plated in DMEM with and without induction (2 ng/μl dox, Sigma) in a 48-well plate grown first in the incubator. Two days post induction for HEK293TLP-mCherry cells or three days post induction for HEK293TLP-BFP cells, the plate was removed from the incubator, DMEM from half of the samples was gently removed and replaced with clear FluoroBrite DMEM (Gibco A1896701), and the plate was placed into the microplate reader to measure fluorescence levels [EBFP ($\lambda_{ex} = 381$, $\lambda_{em} = 445$) and mCherry ($\lambda_{ex} = 565$, $\lambda_{em} = 610$)]. The plate was then removed from the microplate reader and cells were counted using a Nucleocounter NC-250™ (Chemometec). To follow EBFP expression over time, HEK293TLP-EBFP cells were plated into a 48-well plate according to the plate reader assay and treated with 0 to 2 ng/μl dox (Sigma) to induce EBFP expression. The plate was then placed into the microplate reader and the assay started. Every six hour, following $Abs_{430}$ and $Abs_{560}$ measurements, EBFP fluorescence was also recorded for seven days.

## Statistical analysis

A two-sided t-test was used to assess the statistical significance of $CF$ resulting from different growing conditions, $\mu_{cp}$, and fluorescent proteins expression levels.

## Reporting summary

Further information on research design is available in the Nature Portfolio Reporting Summary linked to this article.

## Data availability

All source data for the data sets and Fig.s presented in the manuscript are available in the source data file published alongside this manuscript. Constructs are available upon request to the corresponding authors. Plasmid maps of integrated constructs are available within the source data file. Source data are provided with this paper.

## Code availability

The code used for data analysis is freely available as a Supplementary software file and data analysis is described in Supplementary Note 1.

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

## Acknowledgements

This work was supported by the New Investigator Award no. WT102944 from the Wellcome Trust U.K. (WT102944 to A.G. and M.I.) and the Biotechnology and Biological Sciences Research Council (grant BB/V00882X/1 to A.G and F.C.). F.C., C.E.B. and C.B. acknowledge the support of the Royal Society International Exchanges 2018 Round 1 grant IES \R1\180027. The authors would like to thank *Matreyek* et al. for providing the HEK293T-LP cell line adopted in this work, Paul Freemont and Jose Jimenez for feedback on the manuscript.

## Author contributions

A.G., M.I. and F.C. conceptualised the research. A.G., R.D.B., D.P., M.S. and Z.Y. performed the measurements. C.E.B. and C.B. performed modelling and mathematical analysis. A.G. and C.E.B. analysed the data. F.C., M.I. and C.B. contributed funding. A.G., C.E.B., C.B., M.I. and F.C. wrote the manuscript. All authors read and edited the manuscript.

## Competing interests

The authors declare no competing interests.
