## [Peer Review File · Nature Communications]

Reviewers' Comments:

Reviewer #1:

Remarks to the Author:

The authors have developed a correlation between a pH-sensitive dye and cell number as a means for non-invasive cell counting. The studies confirm such a relationship with basic studies of cell growth in non-engineered cells and engineered cell lines as well as drug testing. However, the data shows that this phenomena is cell type dependent and media dependent thereby making it less generalizable. Furthermore, there can be cells that do not growth but simply impact pH that would lead to false positive results which is a major flaw in the technique. In all, this reviewer is not enthusiastic that this method will be embraced by the mammalian cell culture community.

Reviewer #2:

Remarks to the Author:

It was a pleasure reading this manuscript. High throughput yet relatively accessible mammalian cell growth assessment methods are indeed in demand, and the authors have come up with a relatively simple and elegant solution.

In this manuscript, a novel high-throughput cell growth assay was developed. The assay is based on the change in phenol red absorbance in the growth media throughout cell growth. The authors compared the assay to the conventional cell counter assays in order to assess the reliability and validity of the method, and demonstrated several applications of such method, such as monitoring cell growth in difference conditions (temperature and sugar source), cell death assay post doxorubicin treatment, and characterisation of a synthetic construct and its expression in an engineered cell line. Authors used both suspension and adherent cell lines.

The manuscript is well-written, clear and concise. The conclusions are backed. The methodology is clear and complete. It would be great to see a web-based tool or plugin to derive quantitative information from the plate reader data in the future.

I have several comments in regards to the experiments the authors performed:

1. When growing suspension cells, you only shake them every 6 hours prior to taking measurements. Would you be able to comment on why you didn't grow them in the shaking incubator while only using a plate reader for the measurements?
2. For doxorubicin assay, you didn't take cell counter measurements as a comparison. Would you be able to comment on why? I assume that the dying cells would additionally affect media pH and it would be interesting to see how that would affect CF, linearity and the resulting quantitative information on cell growth.
3. In both Doxo assay and BFP expression experiments, you didn't provide the background readings, i.e. cells with the addition of the drug solvent (e.g., DMSO only, if Doxo and Dox were dissolved in DMSO). Would you be able to comment on that, please?
4. Would you be able to expand the discussion section by considering the limitations of this method and consider possible solutions to those limitations? For example, a couple of limitations I see: (i) does the absorbance of phenol red limit the use of certain fluorescent proteins in construct characterisation assays? (ii) are there any other reagents/components commonly used in cell culture that would affect the absorbance readings?
5. Lastly, the following sentence in the Doxo section is a bit confusing and required editing (displaying): By contrast, K562 cells displayed less sensitivity to doxo treatment, displaying a constant GI and μp (Figure 3a and 3b bottom) with cells still displaying 90 to 100% μp , in the presence of doxo concentrations up to 500nM (Figure 3b).

Unfortunately, I don't have much expertise to comment on the mathematical or coding aspect of this research.

To sum up, the authors have done great work in developing the assay. I would recommend this manuscript for publication after the comments have been addressed.

Reviewer #3:

Remarks to the Author:

This manuscript uses the change in absorbance ratio of a media additive to provide a measure of cell growth. This is a potentially important new tool for high-throughput mammalian cell assays. However, I believe there are significant issues with the manuscript that need to be addressed prior to publication.

Major comments:

Does $GI = 0$ correspond to $C = 0$ which should be 0 cells? If not, this should be part of the normalisation (see later comment about problems with CF). If so, the linear model should not be fitting an intercept so the model should be $C = GI * m \rightarrow \ln(C) = \ln(GI) + \ln(m)$.

The methods outlined in Supp note 1 are ambiguously described and possibly biased.

- The algorithm detailed in (i) should be written following the conventions for describing algorithms i.e. using pseudo-code, rather than in prose. It is currently quite hard to follow.
- If my understanding is correct, the algorithm finds an upper limit (with an arbitrary set of lower limits) for the linear region in the data first, then a lower limit, by maximising the R-squared of a linear model. This is rather unconventional and possibly introduces bias. A better and more rigorous method would be to include the lower and upper data limits as parameters to optimise in the regression.

"An automatic and analytic procedure to determine the exponential phase of growth is the one we previously adopted in Enrico Bena et al1, where the growth curves were fitted by using a logistic function whose parameters were the lag time, the exponential growth rate and the level of saturation. The edges of the exponential phase of growth were analytically determined from the logistic function.": You have already stated that the two methods are only comparable during exponential growth. The you fit a model to the full timeseries, note that you struggled to do this, and then did it manually. A simpler, and perhaps more robust method, would be to find the steepest gradient of the growth curve on a log scale.

"We found that CF for K562 cells is around 1, suggesting that the indirect μ computed through GI is a proxy for the effective μ of the population obtained by direct cell counting": If GI is proportional to C (which it should be) then the CF should always be 1. A CF not equal to 1 (taking into account error) indicates that the normalisation hasn't been performed correctly. I think, therefore, that this CF parameter is only useful as a check for correctness and should not be viewed as something that one aims to calculate. The useful parameter for this work is the proportionality constant $C = GI * m$, as this allows one to convert between the arbitrary units of GI to the actual number of cells.

Can you compare the accuracy of your method to the accuracy of absorbance measurements to predict cell count for bacteria? It is not obvious whether a ~90% accuracy in C prediction from GI is particularly good.

Minor comments:

I would argue that, given that you are using C as your robust measure of cell number, this should be your "explanatory" variable (x-axis) in your linear regression – and all of your correlation plots – rather than GI.

What is the argument for having different boundaries of linearity for different cell types?

"C0 and GI0 are values of C at zero hour and GI at six hours respectively": why is GI0 at 6 hours?

"For plate reader measurements, we discarded the value at zero hour since it corresponded to an outlier value.": Is this for all experiments? Why is you 0 hour measurement an outlier?

Fig 1C (and all absorbance + counting curves): you should try scaling the secondary y-axis with the model calculated scaling parameter. Currently fig1C looks like there is significant lag in the two

measurements but I actually think it is due to the scaling.

Fig 3: in the bottom left panel, why are growth curves diverging before the doxo is added?

Please give speed and amplitude of shaking in the plate reader.

Point-by-point responses to reviewer comments

Reviewer #1 Remarks to authors: *The authors have developed a correlation between a pH-sensitive dye and cell number as a means for non-invasive cell counting. The studies confirm such a relationship with basic studies of cell growth in non-engineered cells and engineered cell lines as well as drug testing. However, the data shows that this phenomenon is cell type dependent and media dependent thereby making it less generalizable.*

We would like to thank the reviewer for taking the time to read and assess our manuscript. Our data show that a linear relation between $\ln(GI)$ and $\ln(C)$ is always present in the exponential phase of growth for the five cell lines we have considered, including both adherent and suspension ones. What changes, when we changed cell line, is the conversion factor CF , i.e. the ratio between μ_p and μ_c . As we describe in the manuscript, once a cell line is characterised in each experimental condition and CF identified, it is possible to adopt our assay for growth characterisation. A similar approach was recently suggested also for bacteria growth tracking by OD measures¹, which is the approach most widely adopted for tracking bacterial growth over time. For this as well as for other methods routinely adopted in cell and molecular biology (e.g. protein quantification) a calibration curve is needed similarly to what we propose for our assay. The key feature of our method is that a linear relation can always be identified between $\ln(GI)$ and $\ln(C)$ across the cell lines considered confirming that the method is generalisable.

To further corroborate this, we considered HT1080 cells growing in DMEM medium. We purchased DMEM without phenol red to be able to control phenol red concentration within the medium and modify the experimental conditions of our growth assay. We grew the cells in two conditions, in DMEM supplemented with 15mg/L phenol red, as in the original DMEM formulation adopted in the experiments presented in the manuscript thus representing our reference, and in DMEM supplemented with 5mg/L of phenol red, same concentration present in other commercially available media, like the RPMI adopted to grow the suspension cell lines of this study. Results are reported in the figure below, now Figure S4 in the manuscript.

Regarding cell counts, cell growth is not affected by phenol red concentration (Figure S4a). When considering GI , we confirmed that in the presence of 15mg/L phenol red, GI saturates at a higher value and with a higher average growth rate when compared to the case of 5mg/L (Figure S4b-e). Linearity is not impacted ($r^2 = 0.91$ for 15mg/l and 0.95 for 5mg/l) and a corresponding CF can be identified, similar to what previously shown in Figure 1e (Figure S4f-g). CF for HT1080 growing with 15mg/L phenol red is consistent with the one previously obtained in Figure 1e. For HT1080 growing with 5mg/L phenol red the average CF drops to a range compatible with what previously observed for suspension cell lines grown in RPMI. This additional data set suggests that CF may indeed depend on the experimental condition, and possibly the cell line, adopted in the assay. Nonetheless, it also confirms that a linear relation can be found between $\ln(GI)$ and $\ln(C)$ when the cell line is initially characterised, confirming the generalisability of the workflow. We would also point the reviewer to our reply to Reviewer 3, below, that touches on the same topic.

We have now modified the text in the main manuscript at line 106:

“To test this experimentally, we purchased DMEM where no phenol red is present and considered HT1080 cells growing in DMEM supplemented with 15mg/L phenol red for reference, as in the original DMEM formulation adopted in the experiments presented in this manuscript and in DMEM supplemented with 5mg/L of phenol red, same concentration present in other commercially available media, like the RPMI adopted to grow the suspension cell lines of this study. Cell growth was not

affected by phenol red concentration as displayed by the cell counts (Figure S4a). When considering GI, we confirmed that in the presence of 15mg/L phenol red, GI saturates at a higher value and with a higher average growth rate when compared to the case of 5mg/L (Figure S4b-e). Linearity is not impacted ($r^2 = 0.91$ for 15mg/L and 0.95 for 5mg/L) and a corresponding CF can be identified, similar to what previously shown in Figure 1e (Figure S4f-g). CF for HT1080 growing with 15mg/L phenol red is consistent with the one previously obtained in Figure 1e. For HT1080 growing with 5mg/L phenol red the average CF drops to a range compatible with what previously observed for suspension cell lines grown in RPMI. This additional data set suggests that CF may depend on the experimental condition, and possibly the cell line, adopted in the assay. Nonetheless, they also confirm that a linear relation can be found between $\ln(GI)$ and $\ln(C)$ when the cell line is initially characterised.

Overall, we confirmed that, like other methods routinely adopted in cell and molecular biology (e.g. bacterial OD¹, protein quantification), the workflow requires an initial calibration curve for identification of the linear relation between $\ln(GI)$ and $\ln(C)$. Once this is obtained, μ_p can be treated as a direct proxy for the actual cellular μ_c , making the method generalisable for both suspension and adhesion cells.”

Figure S4. Effect of phenol red concentration on GI curves for HT1080 cells. a) Individual growth curves resulting from counting HT1080 cells cultured with 5mg/L (light orange) or 15mg/L (dark orange) phenol red (n=3). **b)** Overtime measure of Abs₄₃₀ (green and yellow lines) and Abs₅₆₀ (purple and pink lines) for HT1080 cells cultured in DMEM containing 5mg/L or 15mg/L phenol red (n=5). **c)** Overtime Abs₄₃₀/Abs₅₆₀ ratio for DMEM background in control wells containing no cells in the presence of 5mg/L (light blue) or 15mg/L (dark blue) phenol red (n=3). **d)** GI profiles over time for HT1080 cells cultured in DMEM containing 5mg/L (light blue) or 15mg/L (dark blue) of phenol red (n=5). **e)** Representative growth curves resulting from phenol red acidification (GI, blue, left vertical axis) and averaged cell counts (orange, right vertical axis) of HT1080 cells cultured with 5mg/L (light blue and light orange) or 15mg/L (dark blue and dark orange) of phenol red (n=3). Numbers of biological repeats for each sample are reported in Related File 1. **f)** Linear relation between GI and C for HT1080 cells when cultured in DMEM containing 5mg/L (left) or 15mg/L (right). The red line is the best fit of data within the linear region. Coefficient of determinations are indicated ($r^2=0.914$ for 5mg/L and $r^2=0.953$ for 15mg/L). **e)** Bar plot of CF for HT1080 cells cultured in DMEM containing either 5mg/L or 15mg/L phenol red. The height of the bar represents the mean value of the single replicates shown as black dots. Growth rate and CF values are reported in Related file 2 while all replicates can be found in Related File 1.

Furthermore, there can be cells that do not grow but simply impact pH that would lead to false positive results which is a major flaw in the technique. In all, this reviewer is not enthusiastic that this method will be embraced by the mammalian cell culture community

We thank the reviewer for the observation. As also indicated by the manuscript title, our proposed method is aimed at the tracking and characterising mammalian cell growth. We indeed agree with the reviewer that for non-growing cells, like primary cell lines, the method would not be suitable. However, this is clearly out of the scope of the assay, whose aim is to provide a relatively low cost, easy and simple way to automatically follow and assess cellular growth in mammalian systems, something not available at the moment and that, as pointed out by the other reviewers, can find application in routine mammalian cell work and synthetic biology as well.

To take the reviewer's comment into account, and add to our characterisation, we decided to demonstrate that the assay is able to capture changes in cell growth over time and to inform about a change in growth during the assay.

To do this, we performed our assay with Jurkat cells in the absence or presence of colchicine, a widely adopted drug with anti-mitotic activity that is adopted for cell cycle synchronisation². Colchicine does this by inhibiting microtubule polymerization, preventing mitosis and thus pausing cell growth. We reasoned this could represent a useful system to inhibit cell growth and assess the effect on GI in our cultures. We considered GI and C for Jurkat cells growing in the presence (treated) or in the absence (untreated) of colchicine, added at 24 hours after starting the plate reader run. Results are shown in panel a of the figure below (now Figure S8 in the manuscript):

Figure S8. Jurkat cells treated with colchicine. Analysis of growth rate changes in Jurkat cells untreated (0ug/ml) or treated (0.025ug/ml) with colchicine added at 24h after the start of the assay (black arrow). **a)** Growth curves resulting from phenol red acidification (GI, blue, left vertical axis) and cell counts (orange, right vertical axis) of Jurkat cells cultured with colchicine 0ug/ml (dark blue and dark orange) or 0.025ug/ml (light blue and light orange) (n=4). **b)** Histograms of relative μ_c (orange, left) and μ_p (blue, right) values of Jurkat treated cells. Data show μ_c and μ_p for the 0-24h time window (preceding colchicine addition and for the 24-48h time window (after colchicine addition). Data were normalised to 100% growth rate in the 0-24h window. The height of the bar represents the mean value of the single replicates shown as dots. After colchicine addition, μ_c is decreased by ~100%, and μ_p is decreased by ~75%.

For both untreated and treated cells, we calculated the growth rates (μ_c and μ_p) in the first 24 hours (i.e. prior to drug administration to the treated cells, black arrow) and compared to the growth rates in the following 24 hours (i.e. soon after drug administration for treated cells while these cells are not dividing but cell number is not decreasing as indicated by the cell counts).

In the first 24 hours, the cells are all untreated and grow as expected with similar growth rates for both μ_c and μ_p (Figure S8). However, in the following 24 hours, as soon as colchicine is added, cell counts of treated cells stop increasing, as indicated by a $\sim 100\%$ drop of normalised μ_c to nearly ~ 0 (Figure S8b). This drop in μ_c is reflected by a $\sim 75\%$ drop of μ_p to $\sim 25\%$.

This reflective change of μ_p indicates that GI curves can capture changes in growth rate occurring over the duration of the assay.

In conclusions, while we agree with the reviewer that non-growing cells, like primary cell lines, are not suitable for our proposed method, we also report here that this method is able to capture changes in growth rate. Finally, the initial calibration of each cell line, that we suggest performing with GI and C measurements done in parallel, would highlight that the cell line of interest is non-growing and thus unsuitable for this method of cell growth characterisation.

We have now added a paragraph in the manuscript where we consider this aspect (line 193):

“To corroborate these results, we considered Jurkat cells treated with the microtubule inhibitor colchicine, an inhibitor of cell division that, similarly to doxo, impacts cell growth². GI measurement and parallel counting were performed for cells treated and untreated with 0.025ug/ml colchicine 24 hours after the start of the assay (Figure S8). Results showed that after colchicine addition, μ_c decreased by $\sim 100\%$ in treated cells compared to cell growth rate before treatment, while μ_p decreased by $\sim 75\%$ (Figure S8). This reflective change of μ_p indicated that GI curves can again capture changes in growth rate occurring over the duration of the assay thus mirroring the cell counts. To conclude, while our proposed assay is mainly suitable for actively growing cells for which a linearity window can easily be identified, it still can be adopted for semi-quantitative characterisation of the dose-response drug effect on mammalian cell viability with a throughput difficult to achieve with standard cell counting approaches”.

Reviewer #2 (Remarks to the Author):

It was a pleasure reading this manuscript. High throughput yet relatively accessible mammalian cell growth assessment methods are indeed in demand, and the authors have come up with a relatively simple and elegant solution. In this manuscript, a novel high-throughput cell growth assay was developed. The assay is based on the change in phenol red absorbance in the growth media throughout cell growth. The authors compared the assay to the conventional cell counter assays in order to assess the reliability and validity of the method, and demonstrated several applications of such method, such as monitoring cell growth in difference conditions (temperature and sugar source), cell death assay post doxorubicin treatment, and characterisation of a synthetic construct and its expression in an engineered cell line. Authors used both suspension and adherent cell lines.

The manuscript is well-written, clear and concise. The conclusions are backed. The methodology is clear and complete. It would be great to see a web-based tool or plugin to derive quantitative information from the plate reader data in the future. (...)

To sum up, the authors have done great work in developing the assay. I would recommend this manuscript for publication after the comments have been addressed.

We would like to thank the reviewer for recognising that our proposed method is needed, elegant and accessible and that our manuscript is well-written, with backed conclusions, clear and complete.

We also thank the reviewer for the recognition of our work and for suggesting publication.

I have several comments in regard to the experiments the authors performed: 1. When growing suspension cells, you only shake them every 6 hours prior to taking measurements.

We thank the reviewer for the question that allows us to explain our protocol better. The cell lines were purchased from ATCC and from their protocol requirements these cells do not require constant shaking for optimal growth. Thus, these cells are usually cultured in normal flasks without shaking. We added the shaking at fixed 6-hour intervals to homogenise the culture prior to acquiring the measurements. This has now been added to the Methods section.

Would you be able to comment on why you didn't grow them in the shaking incubator while only using a plate reader for the measurements?

The two adopted suspension cell lines do not require shaking as per ATCC guidelines. Performing the assay as the reviewer suggests, by growing the cells in the incubator and using the plate reader only to measure absorbance, would, in our opinion, annul the purpose of the assay which is to have a hand-free, automated assay to follow cellular growth. Indeed, the proposed approach would entail the user to be present to manually move the plate from the incubator to the plate reader at each time point every 6 hours, not surpassing current and more traditional counting methods.

2. For doxorubicin assay, you didn't take cell counter measurements as a comparison. Would you be able to comment on why? I assume that the dying cells would additionally affect media pH and it would be interesting to see how that would affect CF, linearity and the resulting quantitative information on cell growth.

We would like to thank the reviewer for the comment. We initially did not count the cells for this assay to showcase the power of the workflow that enables mammalian cell high throughput screening, with an ease that would not be possible at such resolution with standard counting approaches.

However, to take the reviewer's comment into account, we have now performed new experiments; this time we paired plate reader measures with parallel cell counts, as suggested by the reviewer, so as to analyse the impact of the doxo treatment on GI and linearity.

Both HT1080 and K562 cells were treated with 0, 10, 25, 50, 75, 100, 500, 750 nM and 1 μ M Doxo, added at 24 hours after the start of the assay, as previously done for the experiments presented in the manuscript. At least two wells per condition were considered in each run, with GI measurements and parallel counting. The new GI profiles and corresponding cell counts are shown in the figure below (that is now the new Figure 3 in the manuscript):

Figure 3. Growth of mammalian cell lines treated with doxorubicin can be characterised by a plate reader assay. Response of HT1080 (a) and K562 (b) cells to treatment with increasing doxo concentrations (0nM to 1 μ M). Doxo was added 24 hours after the start of the assay (arrow). Relative μ_p (blue) and μ_c (orange) values of HT1080 (c) and K562 (d) cells treated with 0nM-1 μ M doxo (axis in log scale) were normalised to the average value of μ_p and μ_c for cells treated with 0nM doxo. Dashed blue and orange lines indicate the variation trend of μ_p and μ_c respectively between 10nM and 1 μ M. Equation and r^2 values of these trendline are indicated. Number of biological repeats for each sample are reported in Table S3. Data analysis is described in the Methods section and in Supplementary Note 1.

We confirmed μ_p and μ_c displayed a similar decreasing trend for increasing doxo concentrations (Figure 3c and 3d). This was mirrored by similar sensitivity displayed by GI and cell counts as shown by the regression lines in Figure 3c and 3d with angular coefficients of -15.887 and -16.414 for μ_p and μ_c in HT1080, respectively, and -20.212 and -19.296 for μ_p and μ_c in K562, respectively.

It was possible to identify the specific cell line response to doxo treatment. Indeed, HT1080 cells treated with 75nM doxo resulted in an average 30% decrease of μ_p and 84% decrease of μ_c , whereas K562 cells displayed less sensitivity to the same doxo treatment with an average 17% decrease of μ_p and 33% decrease of μ_c compared to the untreated control. Finally, while analysing the data, we noticed that, for increasing concentrations of doxo, both GI and cell count curves flatten, as a reflection of cell death (see Figure 3 and Related file 1). This led to a more difficult estimation of μ_p and μ_c , due to a less pronounced exponential phase of growth needed to estimate both the linearity region and growth rates (see Supplementary Note 1 and Methods). As previously shown for bacteria, treatment with molecules that affect cell viability may impact such linearity. This needs to be considered as it may lead to the case where the assay is qualitatively very useful but is only semi-quantitative.

We have now added this paragraph to the manuscript at line 175.

3. In both Doxo assay and BFP expression experiments, you didn't provide the background readings, i.e. cells with the addition of the drug solvent (e.g., DMSO only, if Doxo and Dox were dissolved in DMSO). Would you be able to comment on that, please?

We would like to thank the reviewer for the comment. We would like to point out that doxo and dox are resuspended in water, and that no DMSO is present, and that the reference with no induction is the control treated with the solvent (i.e. water in this case).

4. Would you be able to expand the discussion section by considering the limitations of this method and consider possible solutions to those limitations? For example, a couple of limitations I see: (i) does the absorbance of phenol red limit the use of certain fluorescent proteins in construct characterisation assays?

We thank the reviewer for the requested additional experiments which give us the chance to expand on the characterisation of our workflow. In response to point (i): we performed fluorescence measurements at the plate reader for HEK293LP cells hosting integrated landing pads bearing inducible fluorescent reporter expression. We selected two different lines, one source from literature and expressing an EBFP³ and the other developed in house and expressing mCherry. The cell lines were named HEK293LP-BFP and HEK293LP-mCherry and the assay is now described in the Methods section.

We considered how fluorescence per cell changes if phenol red is present or absent from the medium. Results are reported in the figure below (now Figure S10 in the manuscript). The presence of phenol red does not appear to significantly impact the fluorescence readout per cell.

Figure S10. Fluorescence readout for mCherry and EBFP in presence and absence of phenol red in the medium. Cells were grown in the incubator with and without induction with 2ng/ μ l dox for 2 days (a) (mCherry, n=3) and 3 days (b) (EBFP, n=2). Cells were then resuspended in medium with (DMEM) and without (FluoroBrite) phenol red and fluorescence was measured at the plate reader. Cell counts were performed in parallel. A two-sided t-test indicated that average fluorescence level in DMEM and FluoroBrite are compatible (ns, non-significant: $p > 0.05$). Data and statistical analysis are described in the Methods section. Raw values are reported in the source data file.

As suggested, we have now added a paragraph in the manuscript to take this into account (line 224):

“We then reasoned that the presence of phenol red could impact fluorescence readout. Thus, before assessing the fluorescence response of HEK293LP-EBFP, we assessed if crosstalk could be detected between phenol red and fluorescence readings. In order to do so, we expanded the analysis to a second cell line, HEK293TLP-mCherry, developed *in house*. Similarly to HEK293LP-EBFP, HEK293TLP-mCherry codes for an mCherry protein under the control of dox induction (see EBFP diagram in Figure 4a and Source data for sequence). By assessing fluorescence per cell in presence and absence of phenol red, we confirmed that the presence of phenol red in the medium does not impact the fluorescence readout (Figure S10)”.

We also agree if the reviewer that adding on considerations about the limitations of the methods would improve the manuscript and we have thus now added this throughout the text in different sections to provide deeper characterisation of the workflow. We comment on this in the discussion:

“However, when chemical agents are adopted that impact cell growth, the linearity between $\ln(GI)$ and $\ln(C)$ may be affected as previously reported for bacterial OD. While this may impact the quantitative power of the approach, we showed that GI still closely follows the trend of cell counts, providing a useful workflow for high throughput analysis of cell viability”

“This will need to account for an initial screen to ensure no crosstalk is present between a given fluorescent protein and phenol red in the medium, as shown by our data for EBFP and mCherry”

(ii) are there any other reagents/components commonly used in cell culture that would affect the absorbance readings?

We considered a number of chemicals such as the ones used in this study (doxo and dox) but also reagents (DMSO, Doxorubicin), inducers (IPTG, Doxycycline) and selection markers of cell engineering (Puromycin, Hygromycin, Blastidicin) that are commonly adopted in mammalian cell culture. Thus, we followed over time and compared the GI profiles for DMEM and RPMI in the absence or presence of these chemicals. Results are reported in the figures below (now Figure S7 in the manuscript):

Figure S7. Compounds effect on media acidification. Average Abs_{430}/Abs_{560} ratios and error of the mean over time of RPMI (left) and DMEM (right) medium supplemented with either 1uM doxo, 2ng/ μ l dox, 1mM IPTG, 0.5ug/ml Blast, 1ug/ml Puro, 50ug/ml HygB or 0.1% DMSO at T0 (n=4).

From our data, it can be observed that addition of the selected chemicals does not impact GI over time, as the medium supplied with the selected chemicals roughly vary within the same range as the medium alone, this being true for both media.

A paragraph covering this has now been added to the main manuscript (line 167):

“To investigate this, we first verified that the addition of chemicals to the culture medium does not affect GI over time. For this purpose, we supplemented RPMI and DMEM with doxo, as well as with other commonly adopted inducer molecules (doxycycline, dox; and IPTG), antibiotics (blasticidin, blast; puromycin, puro; and hygromycin B, HygB) and the drug solvent DMSO. Addition of these compounds had no impact on medium acidification over time as shown in Figure S7”.

*5. Lastly, the following sentence in the Doxo section is a bit confusing and required editing (displaying):
By contrast, K562 cells displayed less sensitivity to doxo treatment, displaying a constant GI and μp (Figure 3a and 3b bottom) with cells still displaying 90 to 100% μp , in the presence of doxo concentrations up to 500nM (Figure 3b).*

A new doxo section is now present in the manuscript to include the cell counts in the data set.

Reviewer #3 (Remarks to the Author): This manuscript uses the change in absorbance ratio of a media additive to provide a measure of cell growth. This is a potentially important new tool for high-throughput mammalian cell assays.

We would like to thank the reviewer for recognising that our proposed method is a potentially important new tool for high throughput mammalian cell assays.

Major comments:

1. The methods outlined in Supp note 1 are ambiguously described and possibly biased. - The algorithm detailed in (i) should be written following the conventions for describing algorithms i.e. using pseudo-code, rather than in prose. It is currently quite hard to follow.

We thank the reviewer for their comments, as they allowed us to improve the presentation of our algorithm. We agree with the reviewer that pseudo-code rather than prose is much better to present it. We supplied the pseudo-code of our algorithm, with the adjustments we did to address all the reviewers' comments, within the Supplementary Material in Supplementary Note 1.

2. If my understanding is correct, the algorithm finds an upper limit (with an arbitrary set of lower limits) for the linear region in the data first, then a lower limit, by maximising the R-squared of a linear model. This is rather unconventional and possibly introduces bias. A better and more rigorous method would be to include the lower and upper data limits as parameters to optimise in the regression.

3. "An automatic and analytic procedure to determine the exponential phase of growth is the one we previously adopted in Enrico Bena et al1, where the growth curves were fitted by using a logistic function whose parameters were the lag time, the exponential growth rate and the level of saturation. The edges of the exponential phase of growth were analytically determined from the logistic function.": You have already stated that the two methods are only comparable during exponential growth. The you fit a model to the full timeseries, note that you struggled to do this, and then did it manually. A simpler, and perhaps more robust method, would be to find the steepest gradient of the growth curve on a log scale.

Minor comments:

1. I would argue that, given that you are using C as your robust measure of cell number, this should be your "explanatory" variable (x-axis) in your linear regression – and all of your correlation plots - rather than GI.

We thank the reviewer for their careful reading of the manuscript and for their comments, as they allowed us to improve our presented work. To keep all major and minor comments into account, by addressing major comments 2-3 and minor comment 1, we changed our procedure for data analysis. We show the details here below, by considering K562 data as example.

Our reasoning started from the first minor comment, pointing to the fact that, in our method, C is used as the robust measurement and should therefore be the "explanatory" independent variable, while GI should be the dependent one. We totally agreed with this suggestion, which is also in agreement with previous works in the literature¹, where the OD for measuring bacteria is compared to the concentration of cells and C is shown on the x-axis.

By looking at the data plotted in this way (C on the x-axis and GI on the y-axis), we noticed that there is a temporal order in the way the data dispose in the plane $\ln(\text{GI})$ vs $\ln(\text{C})$, as depicted in Figure S12a-c. Indeed, as time increases, first both C and GI increase their values covering almost all their range of variability in ~ 50 hours (this is the exponential phase of growth for both C and GI, red dots in Figure S12a-c), then C saturates and decreases while GI only saturates (orange to blue dots). This difference in the behaviour of the late growth phases leads to a “simil-hysteresis” in the relation between $\ln(\text{GI})$ and $\ln(\text{C})$, as shown by Figure S12c. For this reason, our original procedure could not be used anymore: as it relied on finding the best linear region by progressively increasing the x variable, this would now include points clearly outside the exponential phase. Indeed, Figure S12d (left panel) shows the results for the original procedure applied to the data expressed as $\ln(\text{GI})$ vs $\ln(\text{C})$. Note that the r^2 , from the original value of 0.915 now drops down to 0.77.

FIGURE S12. Measuring growth curves with manual counts and plate reader with the new procedure compared to the old one. Growth of K562 cells in Glucose at 37°C measured during time with **a)** manual counts and **b)** plate reader shown in logarithmic scale. Data are binned in time-windows of 50 hours; each color represents one time-window. **c)** y-axis of panel b) ($\ln(\text{GI})$) as a function of the y-axis of panel a) ($\ln(\text{C})$). The data in the first 50 hours (in red) cover the region of fastest growth of curves in panel a) and correspond to the data shown in Figure 1d in the manuscript. **d)** Comparison between the old (on the left) and the new (on the right) procedure of data analysis to obtain the relation between $\ln(\text{GI})$ and $\ln(\text{C})$. Dashed red lines represent the boundaries of the region of linearity, gray dots are the same data as in panel c. Red lines display the fit of the data within the area delimited by the dashed lines, r^2 values of the fits are explicated in the panels. Thus, the changes requested by major comment 2 cannot be actioned when swapping the axis. Based on the observation that the region of linearity corresponds to the region of fastest growth of both C and GI as a function of time (i.e., the exponential phase of growth), we reconsidered our data analysis flow and derived a more straightforward method, which is based on the following points.

- 1) The C growth profiles, expressed as $\ln(C)$ vs time, are analysed and the region of fastest growth, i.e., the exponential phase of growth, is identified automatically (see below for details). The slope of the linear fit of the selected data is μ_C .
- 2) We selected the very same time-window in the corresponding growth profile $\ln(GI)$ vs time. The slope of this second fit is then μ_P .
- 3) Finally, the data selected in previous steps 1) and 2) are fitted to obtain the relation between $\ln(GI)$ and $\ln(C)$ of the form $\ln(GI) = m \cdot \ln(C) + q$, being m and q parameters of the linear fit.

In step 1), the region of fastest growth is obtained as the time-window in which the growth rate is maximum. To do this, i) we performed linear fits of at least 3 consecutive points in sliding windows that start from time 0 and slide until the end of the growth curve (see Table I in Supplementary Note 1). This has been done to include the lower and upper data limits as parameters to optimise in the regression, following reviewer's suggestion. ii) We selected the range of data points whose linear fit gives the maximum slope (steepest gradient). The slope of the linear fit of these data points is the growth rate of the exponential phase (μ_C). This procedure is performed on curves obtained with manual counts and expressed in log scale, normalized to their initial value. The choice of having at least 3 data points has been based on our interest in estimating the average growth rate in the exponential phase and not the maximum growth rate. The latter, given the resolution of our manual counts, could be more easily biased by eventual experimental variability in cell counts. For conditions with longer duration of the exponential phase (as Jurkat growing in Mannose and both K562 and Jurkat at 33°C), we increased the minimum number of data to be fitted to 4 (Jurkat in Mannose) or 5 (growth at 33°C). However, the supplied script for data analysis allows the user to choose the minimum number of data points on which performing the fit.

The linear relation that we found between $\ln(GI)$ and $\ln(C)$ holds within the exponential phase of growth. This allows to directly relate the conversion factor CF , defined as μ_P/μ_C , to the fit parameter m . Indeed, in the exponential phase of growth, it holds:

$$(1) \ln(C/C_0) = a + \mu_C * t \rightarrow \ln(C) = a + \ln(C_0) + \mu_C * t = a' + \mu_C * t;$$

$$(2) \ln(GI/GI_0) = b + \mu_P * t \rightarrow \ln(GI) = b + \ln(GI_0) + \mu_P * t = b' + \mu_P * t;$$

$$(3) \ln(GI) = m * \ln(C) + q .$$

From (1), $t = (\ln(C) - a')/\mu_C$, thus substituting in (2), $\ln(GI) = b' + \mu_P * (\ln(C) - a')/\mu_C$, that is

$\ln(GI) = (b' - a' * \mu_P/\mu_C) + \mu_P/\mu_C * \ln(C)$. From (3) it follows $m = \mu_P/\mu_C$. m is what we defined as the conversion factor CF . Imagine now you want to know the growth rate μ_C^* of a population of cells you are following at the plate reader. It is enough to measure, at the plate reader, its growth rate within the exponential phase and divide it by the parameter m you have found when doing the calibration, $\mu_C^* = \mu_{Pmeasured}/m$.

The compatibility between m and CF is shown in Figure S13a-c where, per each experiment discussed in Figure 1 and 2 of the main paper, the points (CF, m) lie on the bisector.

Figure S13. Relation between CF and m per each individual growth curve. The parameter m (+/- error) obtained from the linear fit $\ln(GI)=m*\ln(C)+q$, and $CF = \mu_p/\mu_C$ for each individual growth curve are scatter plotted in red. In black is the bisector (i.e., $m = CF$). In all cases the bisector passes within the error bars. In **a**) are the conditions analysed in Fig.1 of the main text together with HEK293TLP-EBFP; in **b**) are the growth conditions at 33°C, in **c**) K562 and Jurkat growing in Mannose at 37°C.

The results obtained agree with those obtained with our original data processing, as shown in the Figure and Table below:

Comparison between the old and new procedure. Comparison of the CF obtained with the old (in grey) and new (in red) procedure of data analysis for conditions at 37°C (**a**) and 33°C (**b**). T-tests comparing each condition gave p-values ≥ 0.098 (see Table below).

	CF old	CF new	p-value
K562 Glu 37°C	(1.26 ± 0.05)	(1.24 ± 0.06)	0.80
Jurkat Glu 37°C	(0.98 ± 0.04)	(1.15 ± 0.12)	0.18
HT1080 Glu 37°C	(1.78 ± 0.08)	(1.76 ± 0.10)	0.89
HEK293T Glu 37°C	(2.00 ± 0.12)	(1.89 ± 0.08)	0.52
HEK293T-LP Glu 37°C	(1.51 ± 0.12)	(1.45 ± 0.18)	0.79
K562 Man 37°C	(1.12 ± 0.09)	(1.12 ± 0.10)	0.95
Jurkat Man 37°C	(1.13 ± 0.06)	(0.96 ± 0.08)	0.12
K562 Glu 33°C	(0.97 ± 0.04)	(1.09 ± 0.05)	0.098
Jurkat Glu 33°C	(1.45 ± 0.09)	(1.58 ± 0.18)	0.52
K562 Man 33°C	(1.09 ± 0.04)	(1.10 ± 0.05)	0.88
Jurkat Man 33°C	(1.27 ± 0.09)	(1.20 ± 0.08)	0.60

Average CF values across the different growing conditions, quantified with the old and the new method. The third column shows the two-sided p-values of a Test. The two tests give compatible results ($p \geq 0.098$)

In agreement with what described above, we changed all figures showing the $\ln(GI)$ vs $\ln(C)$ relation, and tables. The code for analysing the data and the Supplementary Note 1 with its description are once again provided as Related file 3 in the manuscript. The main text has been changed accordingly.

Major comments:

4. Does $GI = 0$ correspond to $C = 0$ which should be 0 cells? If not, this should be part of the normalisation (see later comment about problems with CF). If so, the linear model should not be fitting an intercept so the model should be $C = GI * m \rightarrow \ln(C) = \ln(GI) + \ln(m)$

5. *“We found that CF for K562 cells is around 1, suggesting that the indirect μp computed through GI is a proxy for the effective μc of the population obtained by direct cell counting”: If GI is proportional to C (which it should be) then the CF should always be 1. A CF not equal to 1 (taking into account error) indicates that the normalisation hasn't been performed correctly. I think, therefore, that this CF parameter is only useful as a check for correctness and should not be viewed as something that one aims to calculate. The useful parameter for this work is the proportionality constant $C = GI * m$, as this allows one to convert between the arbitrary units of GI to the actual number of cells.*

We thank the reviewer for their careful reading and comments, as they allowed us to improve the presentation of our work. Major comments 4 and 5 are addressed together here below.

First, we want to underline that i) GI data are always subtracted by the background and, ii) to calculate growth rates the data are normalized to their initial value. Thus, when $C=0$, which corresponds to 0 cells, also $GI=0$ (taking background variability into account). Indeed, the relation we show is not $GI = m * C + q$, but $\ln(GI) = m * \ln(C) + q$, from which we derive $GI = \exp(q) * C^m$ (and from here it is clear that when $C=0$ then $GI=0$, as the reviewer was considering, but with the intercept q that may be different than 0).

We would like to underline that there is no reason why the conversion factor CF , i.e., the angular coefficient m when looking at the $\ln(GI)$ vs $\ln(C)$ relation, must be equal to 1. Indeed, as we showed by replying to Reviewer #1, different concentration of Phenol Red within the cell growth medium may alter the absorbance of the medium itself (and thus the $\ln(GI)$ vs time profile) but not the way the cell population evolves in time ($\ln(C)$ vs time). This implies that, within the exponential phase, the slope of the linear fit amongst $\ln(GI)$ vs $\ln(C)$ data may vary. The take home message however is that it is enough to measure the CF to convert the growth rate measured in the exponential phase at the plate reader into a “real” growth rate.

6. Can you compare the accuracy of your method to the accuracy of absorbance measurements to predict cell count for bacteria? It is not obvious whether a ~90% accuracy in C prediction from GI is particularly good.

We would like to thank the reviewer for their comment, which gave us the opportunity to expand the discussion about the validity of our method. We are not fully sure we captured the meaning of the comment as the reviewer intended, and we are thus happy to further clarify in case this is needed.

We reasoned that there are different ways to measure the accuracy of our method. Here below we report two different analyses, both suggesting that the accuracy in predicting cell counts from absorbance/optical density measurements are similar for mammalian/bacterial cells.

1st analysis

A reasonable measurement of the accuracy of our calibration method is the percentage of C effectively measured C_{measured} that falls within the error of the model used to predict C from a given GI measured at the plate reader ($C_{\text{predicted}}$).

We thus used a triplicate of growth curves for K562 cells, measured both at the plate reader and by manual counts, to generate the linear model $\ln(GI) = m \cdot \ln(C) + q$ by fitting all the data within the exponential phase. The fitted parameters m and q come with errors, so that we can estimate the ‘minimal’ ($\ln(GI_{\text{min}}) = (m - \text{err}_m) \cdot \ln(C) + (q - \text{err}_q)$) and ‘maximal’ ($\ln(GI_{\text{max}}) = (m + \text{err}_m) \cdot \ln(C) + (q + \text{err}_q)$) boundaries of our linear model. We then asked if, given a certain GI from a growth curve not belonging to the triplicate used to generate the model, its C_{measured} falls within the error of the model. To do this, we selected only the GI values belonging to the region for which the linear model holds. We did this for GI values belonging to 3 growth curves from which we had both absorbance and manual count measurements. By repeating this procedure with different combination of 6 growth curves, we obtained that the 100% of our measured C_{measured} falls within the model error for the predicted C , $C_{\text{predicted}}$ (see one representative example in Figure S14a). Then, to compare the accuracy of our method to that of optical density (OD) measurements used to predict cell counts for bacteria, we i) generated 6 growth curves for DH10B bacterial cells, followed cells over time both by counting them directly and by measuring the OD of the population, and ii) applied the same workflow for data analysis described above. In case of bacteria, however, it is known that, for given regimes of growth, OD is directly proportional to C^1 , which implies that the linear model used to fit the exponential phase of growth is $OD = m' \cdot C + q'$. Again, the fitted parameters have errors, which allow to define ‘minimal’ and ‘maximal’ boundaries of the model, and in turn to ask if a C measured C_{measured} from a given OD value falls within the error of the model.

We applied this procedure to the 6 growth curves, by using 3 of them for constructing the model and the remaining 3 for testing it. By repeating this procedure for all the possible combinations of the 6 growth curves, our data show that on average $\sim 78\%$ of our measured cell counts fall within the minimal and maximal boundaries of our model for predicted C , $C_{\text{predicted}}$, obtained as a deviation from the best fit of ± 1 error of the parameters of the fit. When the boundaries are considered as twice the error of the fit parameters, on average the $\sim 95\%$ of the data fall within such region (see Figure S14 for a representative example). This analysis suggests that the accuracies of our method applied to mammalian or bacterial cells are similar.

FIGURE S14. First method for accuracy measurement on K562 and DH10B cells. a) Results for K562 cell line. GI as a function of C within the exponential phase of growth: lines are the best fit model (obtained by fitting the data from 3 growth curves) ± 1 error over the parameters determining the minimal and maximal boundaries; dots are data belonging to 3 other growth curves used to test the model. Inset: the same as the main panel but in log-log scale. **b)** Results for DH10B cells. Lines are the best fit model (obtained by fitting the data from 3 growth curves) ± 1 (red) or 2 (purple) errors of the parameters determining the minimal and maximal boundaries; dots are data belonging to 3 other curves used for testing the model. In the shown example, 85% of the data are within ± 1 error of the fit, the 100% is within ± 2 errors of the fit.

2nd analysis

To corroborate the results obtained with the first analysis, we then quantified the accuracy in cell concentration estimates as defined in *Beal et al*⁴ and compared our results to those obtained there for bacteria. In *Beal et al*, the authors defined the accuracy as the ratio between the estimated mean fluorescence per cell (fluorescence from *E. coli* normalized by calibrated OD measurements) and the mean fluorescence per cell measured by calibrated flow cytometry. We reasoned that, in our case, this would mean to compare the mean cell count predicted by our calibration method ($C_{\text{predicted}}$) to the mean cell count effectively measured (C_{measured}), which reduces to the ratio $A = C_{\text{predicted}}/C_{\text{measured}}$. The closer to 1 A is, the more accurate the method is in estimating cell counts. We thus built the linear model using a triplicate of growth curves for which we have both cell counts and absorbances, and then measured the accuracy as the mean A derived from all GI data within the exponential phase for three growth curves, different from those used to build the model. By repeating this procedure for 20

different combinations of growth curves, we found that on average $A = (1.07 \pm 0.02)$ for K562 cells, in agreement with the accuracy of microsphere dilution method investigated for bacterial cells⁴.

This result suggests that the accuracy of our method in estimating cell counts from absorbance measurements is like that evaluated to estimate cell counts from OD in bacteria. Hoping to have correctly interpreted what the reviewer meant by comparing accuracies, we are of course open to discussion.

We added a Supplementary Note 2 in the manuscript to describe the above workflow for estimation of the accuracy of our method with the above description. We have also added a paragraph in the main text (line 243):

“We then compared how accurate the prediction of C from GI is compared to bacterial OD. We compared the mean cell count predicted by our calibration method ($C_{\text{predicted}}$) to the mean cell count effectively measured (C_{measured}), which reduces to the accuracy $A = C_{\text{predicted}}/C_{\text{measured}}$. The closer A is to 1, the more accurate the method is in estimating cell counts. The procedure and results of the accuracy calculation can be found in Supplementary Note 2, where we identify a similar accuracy in our approach compared to the accuracy of OD measurements $A = (1.07 \pm 0.02)$ for K562 cells, in agreement with the accuracy of microsphere dilution method investigated for bacterial cells”.

Minor comments:

2. What is the argument for having different boundaries of linearity for different cell types?

We would like to thank the reviewer for this question, that allowed us to expand on the description of the linearity region, by accounting for differences among cell lines and their growth conditions. The different boundaries of linearity in both C and GI for different cell types reflect differences in i) the saturation level that can be reached by each cell line and ii) the growth medium in which the cells are growing. Indeed, the linearity region corresponds to the exponential phase of growth that, in turn, informs about the range of maximum variability of i) the size of the monitored population of cells (C) and in turn ii) the medium acidification (GI). Concerning the former variable, as it can be noticed by the orange growth curves in Figure 1c,f and Figure 2a-c, and as previously showed in *Enrico Bena et al*⁵, the saturation level reached by different cell types, and therefore the range covered by C during the entire growth curve, is different and depends on the cell line (K562 and Jurkat in this case). Regarding the range of variability of GI, as we showed when addressing Reviewer 1's comments, this can be explained by the different concentrations of phenol red within the growth medium supplied to the cells (Figure S4). Indeed, the GI range of variability of K562 and Jurkat are similar since both cell lines were grown in RPMI medium that, in its standard configuration, is supplied with 5 mg/L of phenol red. Analogously, the range of variability of GI amongst adherent cells are similar, as these cells were grown in standard DMEM medium supplied with 5 mg/L of Phenol Red.

3. “C0 and GI0 are values of C at zero hour and GI at six hours respectively”: why is GI0 at 6 hours?

“For plate reader measurements, we discarded the value at zero hour since it corresponded to an outlier value.”: Is this for all experiments? Why is you 0 hour measurement an outlier?

We thank the reviewer for their comments, which allowed us to improve our presentation.

The measurement GI0 shown in the plots was set to be the second time-point with our experimental resolution (i.e., 6 hours) as the first one always appeared to be an outlier in all the experiments. As can be seen in Figures S1b, S1e, S2c, S2d, S4c, S7 and S9b the first measurement is outside the range of variability of the rest of the curve, and this is the case for both DMEM and RPMI growth media. Similarly, when counting is done in parallel to the plate reader assay, the first measurement after restarting the plate reader is also an outlier that is discarded (as now mentioned in the exclusion data section of the methods).

To highlight this further, we have investigated the absorbance background from RPMI and DMEM medium with an increased time resolution, i.e. measurements every 15 minutes for 20 hours. It takes ~2 hours for the background absorbance values to fit within the variability of the subsequent curves. This is compatible with the choice of having GI0 set at 6 hours for the time resolution of the experiments presented here.

Variation of absorbance background over time in the plate reader assay. Average background and error of the mean for Abs₄₃₀/Abs₅₆₀ ratio of RPMI (blue) DMEM (orange) over time (n=24) in wells containing no cells.

5. Fig 1C (and all absorbance + counting curves): you should try scaling the secondary y-axis with the model calculated scaling parameter. Currently fig1C looks like there is significant lag in the two measurements but I actually think it is due to the scaling.

We thank the reviewer for their comment. The significant lag in the two measurements in Figure 1C is likely due to the different time resolution used for manual counts (24 hours) and plate reader measurements (6 hours). We show here below the same panel as in Figure 1c but with both cell counts and GI with a resolution of 24 hours where the significant lag highlighted by the reviewer is not present. The reason we showed cell counts and GI with different resolution was to show that the workflow allows for closer monitoring of growth overtime but agree that showing the same scaling could have been chosen instead without impacting the strength of the assay.

6. Fig 3: in the bottom left panel, why are growth curves diverging before the doxo is added? We thank the reviewer for their question, as it highlighted an error in graphing this data set for which we apologise. Indeed, this data set is lacking the 24h time point due to a technical mistake and was inappropriately graphed over time. The graph replotted here below is the accurate representation of this data set:

However, due to the lack of 24 hours measurement and subsequent divergence in GI values, curves still appear divergent before addition of doxo. Thus, we have repeated this experiment with parallel counting as per request of Reviewer 2 and now replaced Figure 3 with a new data set.

7. Please give speed and amplitude of shaking in the plate reader.

Shaking is for 20 s with a 2.5 mm amplitude. This has now been added to the Methods section.

References:

- 1 Stevenson, K., McVey, A. F., Clark, I. B. N., Swain, P. S. & Pilizota, T. General calibration of microbial growth in microplate readers. *Sci Rep* **6**, 38828, doi:10.1038/srep38828 (2016).
- 2 Mukhtar, E., Adhami, V. M. & Mukhtar, H. Targeting microtubules by natural agents for cancer therapy. *Mol Cancer Ther* **13**, 275-284, doi:10.1158/1535-7163.MCT-13-0791 (2014).
- 3 Matreyek, K. A., Stephany, J. J. & Fowler, D. M. A platform for functional assessment of large variant libraries in mammalian cells. *Nucleic Acids Res* **45**, e102, doi:10.1093/nar/gkx183 (2017).
- 4 Beal, J. *et al.* Robust estimation of bacterial cell count from optical density. *Commun Biol* **3**, 512, doi:10.1038/s42003-020-01127-5 (2020).
- 5 Enrico Bena, C. *et al.* Initial cell density encodes proliferative potential in cancer cell populations. *Sci Rep* **11**, 6101, doi:10.1038/s41598-021-85406-z (2021).

Reviewers' Comments:

Reviewer #2:

Remarks to the Author:

The submitted and edited manuscript builds up on the previous version and further validates the development of a new methodology for the quantification of mammalian cell growth. This approach uses the pH-dependent media absorbance change readouts as proxy for quantitating the cell growth over time. In general, again, I would like to point out the elegance and comprehensiveness of this approach and would recommend the research for publication.

First of all, I would like to thank the authors for responding to my comments and addressing them thoroughly.

1) Thank you for clarifying the methodology in regards to the suspension cell growth conditions, i.e. Jurkat and K562 cells not requiring continuous shaking. This is now clearly stated in the Methods section.

2) Thank you for your efforts in making additional experiments in regards to assessing the effect of cell media additives on the readout. It led to an important remark that supplementation with any components might impact the reliability of the assay and thus require additional calibrations.

3) Thank you for clarifying the solvent (water) for Doxo and dox.

4) I think the authors did a good job on considering the possible limitations of the method and actually performing additional tests to validate them. It provides important insights into considerations that must be taken into account when implementing the developed assay.

5) The only minor additional suggestions from my side would be to change the word 'basic' in line 68 to, for example, 'foundational'.

To sum up, I am fully satisfied with how the authors addressed my previous comments and improved their work. The methodology is clear and detailed, the experimental pipeline is logical and coherent.

I would recommend this manuscript for publication.

Reviewer #3:

Remarks to the Author:

Response to authors

I want to start by acknowledging and thanking the authors for the considerable work they have done to take on board the comments of myself and the other reviewers and to improve the manuscript.

My understanding of the purpose of this manuscript is to present a new method to allow the estimation of cell number using an easy, high-throughput assay. In simple terms, for such a method to be useful, one needs to be able to calculate a calibration curve of cell count versus some easy measurement and, at a later date, use that calibration curve to determine the cell counts from the easy measurements. My summary of the premise of the manuscript is that:

1. The authors measure cell counts (C) versus 430/560 absorbance (GI)
2. GI is a measure of media pH via Phenol Red colour change
3. Acidification is related to population growth but varies for different cell types and growth conditions
4. The two measures, C and GI, are proportional only during exponential growth and are not linearly proportional

I think this is generally fine – the measurements are easy and quick to undertake, proportionality doesn't have to be linear (though power law is curious), and specificity to cell-type and conditions are common with such experiments.

However, I still have some significant concerns that I outline below.

Suitability of GI

The challenge is that the GI measurement is not similar to those used for other similar calibration methods. For bacterial cell count calibration one can spike in known concentrations of microspheres (or cells) and measure the absorbance. In this manuscript, the measure is of an indirect consequence of cell growth; GI changes due to acidification of the media. Therefore the measurement is not just a measure of the current number of the cells but of how the culture got there. Spiking in a known numbers of cells cannot provide a ground-truth GI measurement as the system has lag. It is probable that growing 1 cell to 1000 cells will produce a different GI measurement from growing 100 cells to 1000 cells. Therefore, the GI measurement itself does not seem to be a robust way to calculate cell counts at a given timepoint.

Demonstration of utility

Regardless of my concerns of the suitability of GI, I feel that the manuscript does not demonstrate a robust and useful method in its current form. From what I can tell, there is no demonstration of the process of producing a calibration curve, using it with new data to predict cell counts, and confirming it is correct (within reasonable error). It is possible that this is being done in Fig S6 and Supplementary Note 2, but as currently presented I can only see the first part demonstrated. Ideally, this should be done with a range of conditions to demonstrate when the calibration curve is reusable and when one needs to generate new calibration data. In particular, one needs to understand how robust the method is to parameters that might be particularly subject to experimental variability such as initial cell count.

Concerns regarding GI vs C relationship model

Writing down the relationship model, as the authors have done in their response, highlights an issue with this method. The authors state (note there are some errors in the Supplementary Material but it is correct in the authors response to reviewers)

$$\ln(GI) = b' + \mu_P t$$

$$\ln(C) = a' + \mu_C t$$

$$\ln(GI) = \left(b' - a' \frac{\mu_P}{\mu_C}\right) + \frac{\mu_P}{\mu_C} \ln(C)$$

$$\frac{\mu_P}{\mu_C} = m$$

$$\ln(GI) = m \ln(C) + q$$

Therefore

$$\ln(GI) = (b' - a'm) + m \ln(C)$$

$$q = (b' - a'm)$$

$$m = \frac{(b' - q)}{a'}$$

Since $b' = \ln(GI(t = 0))$ and $a' = \ln(C(t = 0))$, m depends on the initial cell count and the initial GI measurement. Therefore, the calculated calibration curve and CF is only valid when starting from the same population size, with the same media composition. This may also be why there is such a large variance in the calculated CFs (see for example Fig 1E). I believe this would be corroborated by trying to use the curves calculated here, to predict cell counts from GI curves when starting with different initial population sizes.

In my initial review I suggested that a different model might be more appropriate. That is that you should be fitting $C = GI m$, where C and GI are normalised against the initial timepoint such that $C = C(t) - C(0)$ and $GI = GI(t) - GI(0)$. A simple explanation follows.

The GI measurement should be linearly proportional to the acidity of the media between pH 5-7.5 (i.e. experimental ranges).

Assuming that during the exponential growth phase

$$\frac{dC}{dt} = \mu C$$

$$\frac{d pH}{dt} = k C$$

i.e. that the population of cells, C , increases exponentially with a growth rate μ , and that the rate of change in pH is proportional to the number of cells. Then

$$C(t) - C(0) = (pH(t) - pH(0)) \frac{\mu}{k}$$

$$C = GI m$$

where $m = \frac{\mu}{k}$, or the growth rate of the cells over the acidifying rate per cell.

Reiterating my comments from the first round of review, when fitting on the log-log scale

$$\ln(C) = \ln(GI) + \ln(m)$$

Therefore, the slope should be equal to 1, and deviations indicate error.

This relies on the assumption that the rate of pH change during exponential growth is only proportional to the number of cells in the culture. I am open to the authors disputing this assumption. Indeed the authors say that they showed in response to reviewer 1 that “different concentration of Phenol Red within the cell growth medium may alter the absorbance of the medium itself (and thus the $\ln(GI)$ vs time profile)”. However, I would argue that for the exponential phase in Fig S4D (i.e. < 2 days) the curves of different Phenol red concentrations do match each other.

Variance in CF and m

Regardless of the above comments, the variance in the CFs (see Figs 1E, S6A, S9F) and m (see Fig S13) calculated are so large as to indicate that this is not a particularly robust method to calculate growth properties. The authors carried out extra work to compare the accuracy of their method to a common approach for calibrating bacterial cell count. I’m not sure that I fully understand the two analyses. In the first, the authors use estimated error in fitted parameters, from three samples, to produce minimum and maximum boundaries, and then determine that predicted cell counts from further samples fall within this region. The problem with this approach is that the size of the error region of at least one order of magnitude (possibly more, though it isn’t clear from the plots, Fig S14A). The second analyses, if I understand it correctly, is a very sensible approach. Take three samples from your data to produce a calibration curve, then calculate the sum-of-squared residuals from predictions on other data using that calibration curve. If you do this repeatedly, *at random*, so that calibration curves are eventually constructed from a large subset of the data, you should gain a good picture of the robustness of the method. Unfortunately, the way this analyses is currently presented, it is impossible to determine its rigour. Given the results from the first analyses, and the spread of CFs and m values presented in the manuscript, I do not believe this would produce a comparable value to that for bacterial cell counts.

Summary

I want to make it explicit that I think the authors have clearly undertaken a great deal of extra work following the first round of reviews, and it is greatly appreciated by this reviewer. However, in its current form, I do not believe that this manuscript presents a robust and useful method.

Although, I do not like asking for extra experiments to be performed when reviewing – I do believe that some extra experiments and analyses are important, either to demonstrate the validity and use of your method or to further highlight the need for changes. I would recommend that these experiments should grow cell cultures from different initial conditions (the priority being different initial population counts), calculate calibration curves using one such condition, and try to predict cell counts in the other conditions. This would allow the authors to explicitly demonstrate their methods use in a way that I don’t believe is currently done. It would also allow the authors to perform a proper analyses of the robustness, and therefore the utility of this method.

Point-by-point responses to reviewer comments

Reviewer 2. We would like to thank Reviewer 2 for pointing to “*the elegance and comprehensiveness of this approach and would recommend the research for publication*” and for acknowledging the work we carried out to address all their comments from the first revision round.

Reviewer 3. *I want to start by acknowledging and thanking the authors for the considerable work they have done to take on board the comments of myself and the other reviewers and to improve the manuscript. I want to make it explicit that I think the authors have clearly undertaken a great deal of extra work following the first round of reviews, and it is greatly appreciated by this reviewer.*

We would also like to thank Reviewer 3 for acknowledging the work we carried out to address all their comments from the first revision round. Here below, we would now like to address the remaining comments from the Reviewer.

Variance in CF and m. We would like to start with the comment that points to the variance in CF and m. We refer the reviewer to our previously published work (Enrico Bena et al 2021). We add below a plot taken from the same paper, i.e. Figure 3c. In that work we considered the dependence of mammalian cells growth rate in exponential phase on the initial culture seeding, exactly what the reviewer is asking.

In our work, we found that there exist a range of seedings, around to 10^5 cells/ml (which is exactly the recommended seeding for our cell lines and the one used for experiments presented in our manuscript), for which the variability in the growth rate within the exponential phase (λ_{\max}) is maximum (in the figure each dot represents the maximum growth rate of a population of Jurkat cells (the same holds for K562 cells)). This means that growth curves generated with that initial seeding will be highly variable in terms of maximal growth rate (much more than curves generated with lower seeding N_0 , such as 10^3 cells/ml). If one used a few of these curves to generate the model and obtain the CF, the variability in the growth rates will reflect on the amplitude of the error bar over the CF. This is exactly what we observed, and we show in the plot below that our measured (yellow) and predicted (blue) C values are in line with the previous dataset.

In our published work, we explained the dependence of the growth rate variability on the seeding by considering intercellular communication. Since different seedings lead to different growth rates in exponential phase, we consider that experiments with different initial seeding must be considered as performed in different growth conditions. They will thus require initial characterisation as explained in our manuscript.

We show in the answer below (**Demonstration of utility**) that it is indeed possible to initially characterise growth in a given condition and use this to predict cell counts and growth rates for experiments adopting the same condition, thus performing a calibration to be used for further prediction, as the reviewer requests. However, from the above, we consider that for the seeding more widely adopted in lab-based experiments and advised by commonly adopted manufacturer`s instructions ($\sim 10^5$ cells/ml), there exists an intrinsic variability.

In our work we show how, besides this given intrinsic variability, we are able to measure CF , providing an easy and quick method for mammalian cell growth tracking that is less time consuming and provides higher throughput than traditional cell counting. The price to be paid is a confidence interval higher than that we would have in the case of lower seeding. On the contrary, for lower-seeding experiments, the price to be paid would be a much longer and variable lag time, leading to single growth experiments taking up to 20-30 days, as we previously showed. Please see the plot below (Figure 2a in the published manuscript) displaying variability in the lag time, and thus in the overall experiment timeline as a function of N_0 .

We think the comment of the reviewer is important and have now added to the manuscript an additional supplementary note (now Supplementary note 2) to discuss this.

Suitability of GI. *The challenge is that the GI measurement is not similar to those used for other similar calibration methods.*

We have taken the comment of the reviewer into account and added to our comparison between OD and GI in the text at line 298:

“When comparing our approach to OD, however, it must be considered that, while OD is based on the measurement of scattered light from cells in solution, GI in our workflow is based on media acidification. Thus, GI is a dynamic measurement of mammalian cell growth dependent of the conditions and history of growth, while OD enables assessment of cell concentration at any given moment in time, independently of the history of the cell population”.

The authors carried out extra work to compare the accuracy of their method to a common approach for calibrating bacterial cell count. I'm not sure that I fully understand the two analyses. In the first, the authors use estimated error in fitted parameters, from three samples, to produce minimum and maximum boundaries, and then determine that predicted cell counts from further samples fall within this region. The problem with this approach is that the size of the error region of at least one order of magnitude (possibly more, though it isn't clear from the plots, Fig S14A).

The second analysis, if I understand it correctly, is a very sensible approach. Take three samples from your data to produce a calibration curve, then calculate the sum-of-squared residuals from predictions on other data using that calibration curve. If you do this repeatedly, at random, so that calibration curves are eventually constructed from a large subset of the data, you should gain a good picture of the robustness of the method. Unfortunately, the way this analysis is currently presented, it is impossible to determine its rigour. Given the results from the first analyses, and the spread of CFs and m values presented in the manuscript, I do not believe this would produce a comparable value to that for bacterial cell counts.

We thank the reviewer for the comment. We explained in the previous response that we had not clearly understood what the reviewer had requested. However, in order to address their comment, we decided to adopt the two methods proposed in the old Supplementary note 2 (now Supplementary note 3). We thank the reviewer for clarifying that method 2 is the one more accurately addressing their concern. We took this into account by removing the first method from the Note and leaving only method 2 in it.

First of all, we would like to point the reviewer to the answer above to address their concern regarding the variability.

Secondly, we also took into account the reviewer's suggestion. From the pool of growth curves, we randomly (as the reviewer suggested) selected a set M of 3 curves for constructing the calibration curve, i.e. to infer the quantitative relation between $\ln(GI)$ vs $\ln(C)$. Then, we randomly selected a second set P of 3 curves to predict C ($C_{\text{predicted}}$) and computed the ratio $A = C_{\text{predicted}}/C_{\text{measured}}$. Note that there is no overlap between M and P. We repeated this procedure 20 times, computing each time the mean accuracy A (A_{mean}) per replica. The final value $A = (1.07 \pm 0.02)$ reported in the previous revision, was obtained by the average of the 20 A_{mean} .

Additionally, to corroborate the robustness of our method, we tested the variability of A by changing the size of set M, i.e., the number of curves used for constructing the calibration curve. Specifically, we repeated the above procedures by randomly choosing 2, 3, 5, 10 and 17 curves

for the model and repeated the procedure 100 times per each condition. Figure and Table below (now Figure S14 and Table S6) show that increasing the number of curves used for the model helps in decreasing the dispersion of the accuracy values. However, when considering the coefficient of variation (CV) this is of the order of 0.1, varying from 0.13 (model with 2 curves) to 0.08 (model with 17 curves), suggesting that the variability in *CF* and *A* is intrinsic to the experimental working condition.

Please note that in Figure S14 we reported the mean accuracies obtained for each repeat to highlight the dispersion of the data. We hope that this helps to have a clearer picture of our results.

Figure S14. Comparison of the mean accuracies obtained by constructing the linear model with $N = 2, 3, 5, 10, 17$ curves. Dots are the average of $A = C_{\text{predicted}}/C_{\text{measured}}$ from 3 curves randomly chosen from the pool of curves not used for the model. For each condition, 100 replicates have been run, each time with $N+ 3$ curves randomly chosen from the entire pool of experiments. Bars are the average of the dots. Data are from K562 cells grown in Glu at 37°C.

N curves	A +/- errA	CV
2	1.052 +/- 0.014	0.13
3	1.059 +/- 0.013	0.12
5	1.026 +/- 0.010	0.097
10	1.040 +/- 0.010	0.093
17	1.030 +/- 0.008	0.079

Table S6. Mean accuracy values (*A*), standard error of the mean (*errA*) and Coefficients of variations (*CV*) that is the standard deviation of *A* over its mean, obtained by constructing the linear model with different number of curves.

Overall, these results show that our method is robust in predicting the concentration of cells within the exponential phase of growth of the population independently on the size of set of curves used to construct the calibration curve. Moreover, despite i) the substantial difference in the method used for estimating bacteria (discussed in the response to ***Variance in CF and m***) and our method, and ii) the variability in *CF* and *A* intrinsic of the experimental working condition, the accuracies in estimating the concentration of cells here are similar to the result of the microsphere method reported in *Beal et al*, where they obtained an accuracy of ~ 1.07 .

Demonstration of utility. *There is no demonstration of the process of producing a calibration curve, using it with new data to predict cell counts, and confirming it is correct (within reasonable error). It is possible that this is being done in Fig S6 and Supplementary Note 2, but as currently presented I can only see the first part demonstrated. Ideally, this should be done with a range of conditions to demonstrate when the calibration curve is reusable and when one needs to generate new calibration data.*

We thank the reviewer for their comment. As the reviewer mentions, the requested demonstration was included in Figure S6. where we show the use of a subset of data to compute the average CF and a second subset (not overlapping with the first one) to predict the growth rates. Its validity is shown by the compatibility between the calculated and predicted values of μ_C (Figure S6b) obtained through t-tests.

To take the reviewer's comment into account, we have now repeated the analysis but for the prediction of cell concentration within the exponential phase.

To check its validity, we: i) computed the accuracy ($C_{\text{predicted}}/C_{\text{measured}}$) and check that its value is similar to the one obtained and discussed in Supplementary Note 3, and ii) plotted $C_{\text{predicted}}$ over time and check that their trend matches the trend of C_{measured} vs time (now Figure 1e). For K562 cells cultured in Glu at 37°C, points i) and ii) were satisfied, indeed we obtained accuracy $A = (1.14 \pm 0.06)$ and $C_{\text{predicted}}$ (grey dots in Figure 1e) as a function of time follow C_{measured} (orange dots in Figure 1e), therefore we can conclude that the method supports the estimation of C within the exponential phase of growth.

Results are now part of the new Figure 1, as new panel 1e.

Figure 1e. Concentration of cells ($C_{\text{predicted}}$ in gray and C_{measured} in orange) as a function of time.

We have also added a paragraph to describe this in the text at line 63:

“The slope of the linear fit between $\ln(GI)$ and $\ln(C)$ provides a calibration curve to establish the relation between the actual number of cells and the detected absorbance ratio ($\ln(GI) = m \cdot \ln(C) + q$, Table S2). For K562 cells, a slope value close to 1 (i.e. 0.99) suggests that GI scales almost linearly with C (see Table S2 and Supplementary Note 1). Moreover, from the relation existing between $\ln(GI)$ and $\ln(C)$, C can be estimated from GI as $(GI/\exp(q))^{1/m}$ (Supplementary Note 1). Thus, to fully benchmark our method against cell counts, we decided to use a data subset to establish the relation between C and GI and use it to predict C (C_p) from GI in the remaining data subset in exponential phase (Figure 1e). C_p values overlap with actual C values over time, confirming the relation existing between $\ln(GI)$ and $\ln(C)$ in exponential phase”.

Concerns regarding GI vs C relationship model. *Writing down the relationship model, as the authors have done in their response, highlights an issue with this method. The authors state (note there are some errors in the Supplementary Material but it is correct in the authors response to reviewers)*

We thank the reviewer for their comments and for pointing out some typos within the Supplementary Material, that we have now corrected. Our replies to their comments are below.

$$1) \ln(GI) = b' + \mu_P t$$

$$\ln(C) = a' + \mu_C t$$

$$\ln(GI) = (b' - a' \mu_P/\mu_C) + \mu_P/\mu_C \ln(C)$$

$$\mu_P/\mu_C = m$$

$$\ln(GI) = m \ln(C) + q$$

$$\begin{array}{l} \text{Therefore} \\ q \\ m = (b' - q)/a' \end{array} \quad \begin{array}{l} \ln(GI) \\ = \\ \end{array} \quad = \quad \begin{array}{l} (b' \\ (b' \\ \end{array} \quad - \quad \begin{array}{l} a'm) \\ - \\ \end{array} \quad + \quad \begin{array}{l} m \\ \\ \end{array} \quad \begin{array}{l} \ln(C) \\ a'm) \end{array}$$

Since $b' = \ln(GI(t=0))$ and $a' = \ln(C(t=0))$, m depends on the initial cell count and the initial GI measurement. Therefore, the calculated calibration curve and CF is only valid when starting from the same population size, with the same media composition. This may also be why there is such a large variance in the calculated CFs (see for example Fig 1E). I believe this would be corroborated by trying to use the curves calculated here, to predict cell counts from GI curves when starting with different initial population sizes.

We would like to point out that it is not fully clear to us how the reviewer derived that “ m depends on the initial cell count and the initial GI measurement”, since from their derivation, m depends on a' and b' , but it depends also on q that cancels the dependence on a' and b' giving back the definition of $m = \mu_P/\mu_C$.

However, we agree with them that each condition (in terms of both media composition and population size) should have its own calibration and CF. To support this, we have now added comments to the main text, referring to our previously published work (Enrico Bena et al. 2021), in which we pointed out that different initial seedings should be considered as different initial growth conditions, and give rise to different exponential growth rates. This implies that curves obtained for, say, 10^2 cells/ml cannot be used to predict cell counts from GI curves starting with a seeding of orders of magnitude of difference as, say 10^5 cell/ml. Regarding the large variance in the calculated CF, we refer to what we have written above, in “**Variance in CF and m**”.

2) *In my initial review I suggested that a different model might be more appropriate. That is that you should be fitting $C = GI \cdot m$, where C and GI are normalised against the initial timepoint such that $C = C(t) - C(0)$ and $GI = GI(t) - GI(0)$.*

A simple explanation follows. The GI measurement should be linearly proportional to the acidity of the media between pH 5-7.5 (i.e. experimental ranges). Assuming that during the exponential growth phase

$$dC/dt = \mu C$$

$$d \text{ pH}/dt = k C$$

i.e. that the population of cells, C , increases exponentially with a growth rate μ , and that the rate of change in pH is proportional to the number of cells. Then $C(t) - C(0) = (\text{pH}(t) - \text{pH}(0))/\mu/k$, $C = GI \cdot m$

where $m = \mu/k$, or the growth rate of the cells over the acidifying rate per cell.

Reiterating my comments from the first round of review, when fitting on the log-log scale

$$\ln(C) = \ln(GI) + \ln(m)$$

Therefore, the slope should be equal to 1, and deviations indicate error. This relies on the assumption that the rate of pH change during exponential growth is only proportional to the number of cells in the culture. I am open to the authors disputing this assumption.

We thank the reviewer for their comments, with which however we do not agree. First, we would like to highlight that in our work we do not claim to define a model of growth for our populations of cells, being totally agnostic on the way GI changes with the acidity of the media. We simply observe that, with properly normalized data and in the exponential phase of growth, $\log(GI) = m \cdot \log(C) + q$ (and not $GI = m \cdot C$). Being relations between logarithms, when $C=0$, $\log(C) \rightarrow -\infty$ and the same will do $\log(GI=0)$, independently on the presence of an intercept q or on m different than 1. Indeed, data show that m can be different than 1 and that its value may change with the growth conditions.

3) Indeed the authors say that they showed in response to reviewer 1 that “different concentration of Phenol Red within the cell growth medium may alter the absorbance of the medium itself (and thus the $\ln(GI)$ vs time profile)”. However, I would argue that for the exponential phase in Fig S4D (i.e. < 2 days) the curves of different Phenol red concentrations do match each other.

In Fig. S4d, the exponential phase begins at day ~1 and lasts approximately until ~day 3. Within this interval of time, the curves are very different and, even for time day 1 to day 2, the condition with 15mg/l Phenol Red has systematically higher values than the condition 5 mg/l. We show here below a zoom in of the graph presented in Figure 4d.

Reviewers' Comments:

Reviewer #3:

Remarks to the Author:

I would like to once again thank the authors for their responses. I am satisfied that they have answered my questions and justified their decisions.